# Rendering-Aware Reinforcement Learning for Vector Graphics Generation

**Juan A. Rodriguez**[1,2,3,*], **Haotian Zhang**[5,*], **Abhay Puri**[1], **Aarash Feizi**[1,2,11],
**Rishav Pramanik**[7], **Pascal Wichmann**[6], **Arnab Mondal**[2,8], **Mohammad Reza Samsami**[2,9]
**Rabiul Awal**[1,2], **Perouz Taslakian**[1], **Spandana Gella**[1], **Sai Rajeswar**[1,2]
**David Vazquez**[1], **Christopher Pal**[1,2,4,10], **Marco Pedersoli**[1,2,3]

[1]ServiceNow Research    [2]Mila    [3]ÉTS Montréal    [4]Polytechnique Montréal
[5]Columbia University    [6]Independent Scholar    [7]Stony Brook University    [8]Apple
[9]Google Research    [10]Canada CIFAR AI Chair    [11]McGill University

## Abstract

Scalable Vector Graphics (SVG) offer a powerful format for representing visual designs as interpretable code. Recent advances in vision-language models (VLMs) have enabled high-quality SVG generation by framing the problem as a code generation task and leveraging large-scale pretraining. VLMs are particularly suitable for this task as they capture both global semantics and fine-grained visual patterns, while transferring knowledge across vision, natural language, and code domains. However, existing VLM approaches often struggle to produce faithful and efficient SVGs because they never observe the rendered images during training. Although differentiable rendering for autoregressive SVG code generation remains unavailable, rendered outputs can still be compared to original inputs, enabling evaluative feedback suitable for reinforcement learning (RL). We introduce RLRF (Reinforcement Learning from Rendering Feedback), an RL method that enhances SVG generation in autoregressive VLMs by leveraging feedback from rendered SVG outputs. Given an input image, the model generates SVG roll-outs that are rendered and compared to the original image to compute a reward. This visual fidelity feedback guides the model toward producing more accurate, efficient, and semantically coherent SVGs. RLRF significantly outperforms supervised fine-tuning, addressing common failure modes and enabling precise, high-quality SVG generation with strong structural understanding and generalization.

## 1  Introduction

Generating structured visual code from perceptual inputs, known as the *inverse rendering code generation problem*, aims to translate images or text into executable code that reproduces the target visual content [Rodriguez et al., 2025b, Baulé et al., 2021]. This task has gained momentum with the rise of vision-language models (VLMs) capable of visual-symbolic reasoning and autoregressive sequence generation. Among symbolic formats, *Scalable Vector Graphics (SVG)* [Ferraiolo et al., 2000, Quint, 2003] offer a uniquely expressive target: they are compact, editable, and resolution-independent, and align naturally with the token-based generation processes of language models [Brown et al., 2020]. Recent works successfully frame SVG generation as a code synthesis problem [Wu et al., 2023, Cai et al., 2023, Xing et al., 2024, Zhang et al., 2023, Nishina and Matsui, 2024]. StarVector [Rodriguez et al., 2025b] proposes a model that combines a CLIP vision encoder with a pretrained LLM to directly predict SVG code from images. Other works [Cai et al., 2023, Yang et al., 2025b, Zhang et al., 2023] have explored editing, reasoning, and style control using similar architectures. These models are typically trained using supervised learning on tokenized SVG sequences, achieving strong

---

*Equal contribution

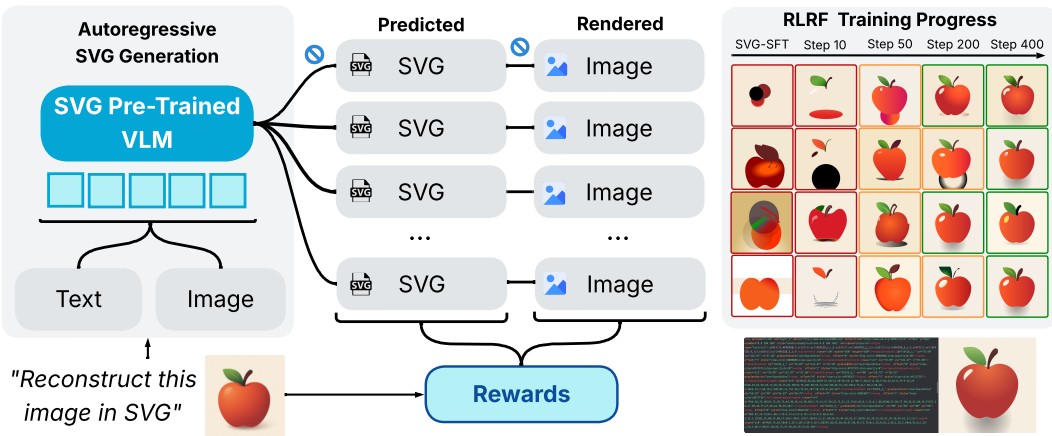

Figure 1: **RLRF Overview.** We present an RL approach for inverse rendering code generation tasks, focused on SVG generation in VLMs. (Left) Given a text or image input, the model generates multiple SVG rollouts, which are rendered and compared to the input to compute rewards based on reconstruction, semantics, and code efficiency. Non-differentiable steps (marked with stop signs) are handled through RL. (Right) A challenging out-of-distribution example with no ground truth SVG. While the base model (SVG-SFT) fails, RLRF enables progressive generalization, producing a meaningful SVG that captures key elements like shadows using gradients.

syntactic and visual accuracy on short examples. However, they suffer from consistency issues over long sequences and often fail to generalize to more diverse or out-of-distribution inputs. We recognize that they operate with a critical limitation: *they do not observe or evaluate the rendered visual output of the code during training*. Token-level losses ensure syntactic correctness but fail to incorporate visual feedback from rendered outputs, causing models to hallucinate, loop, or lose grounding when handling complex inputs (see Figure 1).

This limitation arises from the *non-differentiability of the SVG rendering process* in the context of autoregressive code generation, a challenge that differs from settings addressed by differentiable rasterizers such as DiffVG [Li et al., 2020]. DiffVG makes raster-space losses trainable by expressing an SVG as a small set of continuous primitives, and backpropagating the gradients from the rendered image to those primitive parameters. Autoregressive VLMs, in contrast, emit SVGs one discrete token at a time, making the generation path non-differentiable due to discrete sampling at each decoding step. As gradients cannot flow through discrete sampling steps, differentiable rendering approaches such as DiffVG cannot be applied directly, leaving a critical gap in autoregressive SVG generation.

To address this, we propose **RLRF** (Reinforcement Learning from Rendering Feedback), a method for refining the SVG generation capabilities of pretrained VLMs using feedback from rendered code outputs (images), as depicted in. Figure 1. Our key insight is that rendering, though non-differentiable, can still provide valuable evaluative feedback: rendered outputs can be directly compared to the ground-truth image using automatic and high-quality reward functions. For each input image, the model samples multiple SVG roll-outs, renders them into images, and compares the results to the original.

We design a novel composite reward function integrating complementary reward signals to guide learning: (1) *image reconstruction*, measured using metrics like L2 distance, (2) *semantic similarity*, computed using models such as DreamSim [Fu et al., 2023] or CLIP [Radford et al., 2021]; and (3) *code efficiency*, based on deviations in SVG token length. This hybrid reward setup encourages the model to improve SVG generation along multiple axes. RLRF delivers substantial gains over supervised fine-tuning, addressing common failure modes and enabling models to acquire a deeper understanding of SVG, producing accurate and well-structured vector graphics across diverse inputs.

Our contributions are as follows:

1. We introduce **RLRF**, a reinforcement learning method for SVG generation that leverages rendering rewards to optimize the model, marking the first use of an online RL algorithm for inverse rendering code generation tasks.

2. We introduce a set of **rewards for vector graphics rendering** that combine pixel-level similarity, semantic alignment, and code efficiency to effectively guide models in SVG generation tasks.

3. We demonstrate **state-of-the-art improvements in SVG generalization**, fidelity, and code compactness across tasks and model sizes, supported by extensive analysis.

## 2   Related Work

**SVG Generation**   SVG generation methods typically fall into three categories: classical image processing, latent variable models, and large language model (LLM)-based methods. Traditional methods [Vision Cortex, 2023, Selinger, Peter, 2024, Weber, Martin, 2024] rasterize images by tracing contours and clustering regions, effectively extracting shapes but generating verbose, semantically unstructured code. Latent variable models [Carlier et al., 2020, Wang and Lian, 2021, Ma et al., 2022, Li et al., 2020, Cao et al., 2023] enable better interpolation and style transfer but are bounded by simplified SVG subsets, resulting in less readable code. Recent LLM-based approaches [Rodriguez et al., 2025b] capture the full SVG syntax by treating SVG generation as code generation tasks. Subsequent studies [Zhang et al., 2023, Nishina and Matsui, 2025, Yang et al., 2025b] explored structured generation, editing, and reasoning. However, because these models never validate outputs visually, they often generate inefficient or inaccurate SVG code. To address these limitations, we introduce RLRF, a reinforcement learning-based post-training strategy that incorporates visual feedback into the learning loop, improving visual fidelity, semantic alignment, and code quality.

**Vision-Language Models (VLMs)**   In parallel, vision-language models [Li et al., 2022, Liu et al., 2023, OpenAI, 2023] and code-generating LLMs [Nijkamp et al., 2022, Li et al., 2023b] have advanced significantly, enabling tasks that span both modalities – such as inverse rendering where model generates SVG, TikZ, and CAD code from visual inputs [Belouadi et al., 2024, Rodriguez et al., 2025a, Belouadi et al., 2025, Wang et al., 2025]. However, these models often hallucinate or fail to generalize. Our work addresses this issue by providing visual feedback from the rendered output, which improves both downstream performance and generalization in code-driven visual tasks.

**Reinforcement Learning Post-Training**   Reinforcement learning (RL)[Schulman et al., 2017, Zelikman et al., 2024, Rafailov et al., 2023] has recently emerged as a powerful framework for post-training, particularly in aligning large language models (LLMs) with human or programmatic feedback. GRPO[Shao et al., 2024] has shown that online RL can be both efficient and scalable for LLM alignment. However, most RL applications in code generation remain focused on execution correctness, largely ignoring visual or structural outcomes. For example, CodeRL [Le et al., 2022] uses an actor-critic framework to optimize for functional correctness, while RLEF [Gehring et al., 2024] improves code quality by iteratively refining it based on execution signals. In contrast, our work introduces a composite reward that incorporates rendering feedback, enabling optimization for visual fidelity, semantic alignment, and code efficiency in SVG generation. A more detailed discussion of related work is provided in Appendix D.

## 3   Method

**Autoregressive SVG Code Generation**   Autoregressive VLMs have recently been applied to generate SVG code from images. StarVector [Rodriguez et al., 2025b] trains a VLM by encoding images with a CLIP Vision Transformer [Radford et al., 2021] and projecting features to the language model's dimension. Using next token prediction, it learns to generate SVG code that shows strong semantic understanding and effective primitive identification, leveraging primitives like `<text>`, `<circle>`, and `<polygon>`, and enhancing visuals via color gradients. However, the model struggles with complex images requiring longer SVGs, often hallucinating or looping. This likely stems from a lack of direct visual feedback during training, as it never sees rendered SVGs, limiting accuracy.

**Overcoming the Non-Differentiable Rendering Problem**   In the autoregressive VLM setting, the model learns to translate pixel images into SVG code. However, at inference time, it can easily drift off-distribution, leading to poor visual outputs. Unlike other code generation tasks, inverse rendering tasks like SVG generation offer a unique advantage: the generated code can be rendered into a pixel image and directly compared to the input, enabling precise and inexpensive feedback.

While differentiable SVG renderers such as DiffVG [Li et al., 2020] exist, they rely on latent path representations and are not compatible with our general, token-based approach to code generation. Moreover, because SVG generation involves sampling discrete tokens, gradients cannot be directly propagated. To address this, we propose RLRF (Reinforcement Learning from Rendering Feedback). The model generates SVG code, which is rendered and evaluated using a composite reward that reflects visual fidelity, semantic alignment, and code efficiency. These rewards are used to optimize the model using reinforcement learning techniques [Schulman et al., 2017, Shao et al., 2024], effectively incorporating rendering-based feedback into the training process.

## 3.1 Two-Stage Training

Our training method first adapts the VLM to the SVG domain through *supervised fine-tuning* on the Im2SVG task (SVG-SFT), and then further enhances its SVG generation capabilities using *reinforcement learning* on visual renderings of its own SVG predictions (RLRF).

**Stage1: Supervised Fine-Tuning on Images and SVGs (SVG-SFT)** Let $x_c$ denote the conditioning input (an image or a text prompt) and let $x_s = (x_{s,1}, \ldots, x_{s,L})$ be the ground-truth SVG token sequence. We adapt the base VLM by minimising the negative log-likelihood

$$\mathcal{L}_{SFT}(\theta) = \mathbb{E}_{x_c \sim \mathcal{D}} \left[ -\log p_\theta(x_s \mid x_c) \right] = \mathbb{E}_{x_c \sim \mathcal{D}} \left[ -\sum_{l=1}^{L} \log p_\theta(x_{s,l} \mid x_{s,<l}, x_c) \right]. \quad (1)$$

where $\theta$ are the model parameters. Equation(1) corresponds to teacher forcing; it trains the model to complete *every* prefix in parallel and is therefore highly efficient. After SFT the model can generate plausible SVG code, but it is still unaware of the visual quality of its outputs because no feedback is provided on the rendered image. This model shall be denoted as $p_{\theta_{sft}}(\cdot \mid x_c)$.

**Stage2: Reinforcement Learning from Rendering Feedback (RLRF)** The conditional distribution $p_\theta(\cdot \mid x_c)$ from Equation (1) is subsequently reinterpreted as the stochastic policy during the RL stage. Ground-truth SVG tokens are labeled $x_s$ during SFT, whereas a sampled SVG sequence during RL is denoted $o \sim p_\theta(\cdot \mid x_c)$ to differentiate it from ground truth. After the model learns to *write* SVG code through SFT, we train it to *evaluate* this code by defining a scalar reward $R(x_c, o)$ calculated from the rendered images (details in Section 3.2). For this stage, we want to maximize the objective for a given condition input $x_c$:

$$\mathcal{J}_{\mathrm{RL}}(\theta) = \mathbb{E}_{o \sim p_\theta} \left[ R(x_c, o) \right] - \beta D_{\mathrm{KL}}(p_\theta \| p_{\theta_{sft}}). \quad (2)$$

where $p_{\theta_{sft}}$ is the frozen SFT model and $\beta$ controls the strength of the KL regularization to prevent catastrophic forgetting. This can be optimized by any policy gradient method with a baseline to reduce variance. However, learning a baseline or value network for lengthy, discrete SVG sequences can be costly and unstable. Therefore, we adopt *Group Relative Policy Optimization* (GRPO) [Guo et al., 2025], which eliminates the external baseline by centering rewards within a batch and maintains proximity between successive policies using PPO's [Schulman et al., 2017] clipped-ratio method.

We draw $G$ roll-outs $\{o_1, o_2, \ldots, o_G\}$ conditioned on the same input $x_c$ using the policy $p_{\theta_{old}}(.|x_c)$. The group-centered advantage and probability ratio for roll-out $i$ is:

$$A_i = R(x_c, o_i) - \frac{1}{G} \sum_{j=1}^{G} R(x_c, o_j), \quad \text{and} \quad r_i = \frac{p_\theta(o_i \mid x_c)}{p_{\theta_{old}}(o_i \mid x_c)} \quad (3)$$

Our final GRPO objective to update the current policy $p_\theta$ over the conditioning data is given by:

$$\mathcal{J}_{\mathrm{GRPO}}(\theta) = \mathbb{E}_{x_c \sim \mathcal{D}} \left[ \frac{1}{G} \sum_{i=1}^{G} \min \left( r_i A_i, \, \mathrm{clip}(r_i, \, 1-\epsilon, \, 1+\epsilon) A_i \right) - \beta \, D_{\mathrm{KL}}(p_\theta \| p_{\theta_{sft}}) \right] \quad (4)$$

where $\epsilon$ is a hyperparameter to clip the ratio.

## 3.2 Rewards for Vector Graphics Rendering

We carefully design reward functions that automatically evaluate SVG generation quality across multiple dimensions. *A key advantage of this setup is that we have access to fully automated, high-quality reward signals*, eliminating the need to learn one using human annotations. We focus our

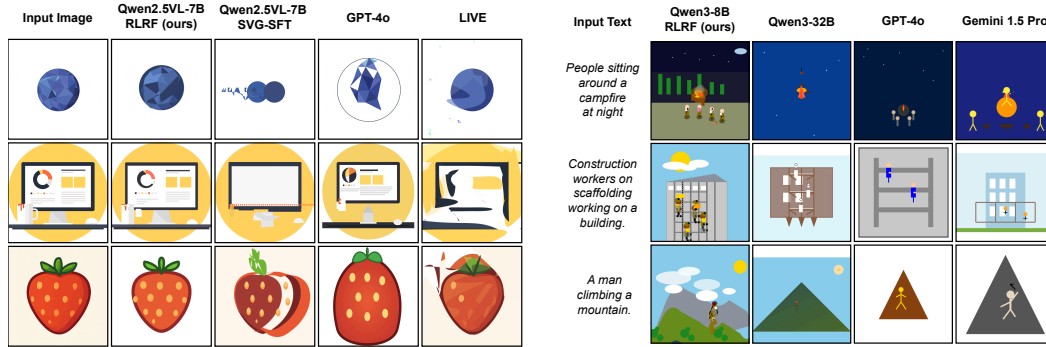

Figure 2: **Im2SVG Reconstruction.** Left: input pixel image. Right: rendered SVG predictions.

Figure 3: **Text2SVG Generation.** Left: input text. Right: generated SVG renderings.

reward design on two core aspects of SVG generation: the visual fidelity of the rendered output and the efficiency of the SVG code in terms of compression. To capture both, we implement a suite of complementary reward functions that assess the rasterized image and the underlying SVG code. These include image reconstruction accuracy, semantic alignment, and code efficiency. By combining these rewards with tunable weights, our framework provides rich and targeted feedback that drives the model to produce SVGs that are both visually faithful and structurally compact. We utilize CairoSVG [Kozea, 2023] as our SVG renderer.

**Image Reconstruction Rewards** To measure how accurately the generated SVG reproduces the input image, we define a pixel-level reward (*L2*) that is both scale-invariant and robust to exposure. Given the input image $I_{\text{in}}$ and rendered prediction $I_{\text{pred}}$, we first normalize both to zero mean and unit variance, then compute the L2 distance and convert it into a clipped reward:

$$R_{\text{img}} = \text{clip}\left(1 - \frac{1}{N}\left\|I_{\text{in}}^{\text{norm}} - I_{\text{pred}}^{\text{norm}}\right\|_2^2, -1, 1\right), \tag{5}$$

where $I^{\text{norm}} = (I - \mu)/\sigma$, and $\text{clip}(x, a, b) = \min(\max(x, a), b)$ bounds the reward within $[-1, 1]$. This normalization ensures the reward emphasizes structural and color discrepancies relevant to vectorization, rather than large uniform regions or exposure artifacts. We also define an edge-aware variant, *L2 Canny*, where we apply a Canny Edge Detector [Canny, 1986] to both $I_{\text{in}}$ and $I_{\text{pred}}$, followed by dilation ($3 \times 3$ kernel, 1 iteration) and Gaussian blur (size 13), which enhances structural alignment based on visual fidelity.

**Semantic Similarity Rewards** For semantic-level rewards, we use *DreamSim* [Fu et al., 2023], which encodes each image using three ViT-B/16 backbones: CLIP, OpenCLIP [Cherti et al., 2023], and DINOv2 [Oquab et al., 2023]. Their feature vectors are concatenated, passed through a linear projection, and compared using cosine similarity. The DreamSim score is computed as $sim = 1 - \cos(\cdot, \cdot) \in [0, 2]$. This provides a strong semantic similarity signal for the Im2SVG task. For shape-focused feedback, we apply *DreamSim Canny*, which uses an edge detector on both prediction and target images before computing DreamSim. A comparison of edge maps emphasizes crisp contours and geometric accuracy while remaining insensitive to variation in color or texture. We convert $sim$ to a reward using $R_{\text{sim}} = 1 - sim \in [-1, 1]$, where higher values indicate stronger semantic alignment. For the Text2SVG task, we use *CLIP* as a reward to compute the cosine similarity between the text prompt and the rendered SVG image in the embedding space. We also utilize *VLM as a judge* to assess generation quality (see details for Text2SVG rewards in Appendix A.2).

**Code Efficiency Rewards.** To encourage compact SVG generation, we define *SVG Length Deviation* as a reward that penalizes excessive token length relative to the ground truth SVG. Let $L_{\text{pred}}$ and $L_{\text{gt}}$ denote the predicted and ground truth SVG token lengths, respectively. The reward is defined as:

$$R_{\text{len}} = 1 - \left(\frac{1}{L_{\text{gt}}}\max\left(0, L_{\text{pred}} - \frac{L_{\text{gt}}}{2}\right)\right)^2, \tag{6}$$

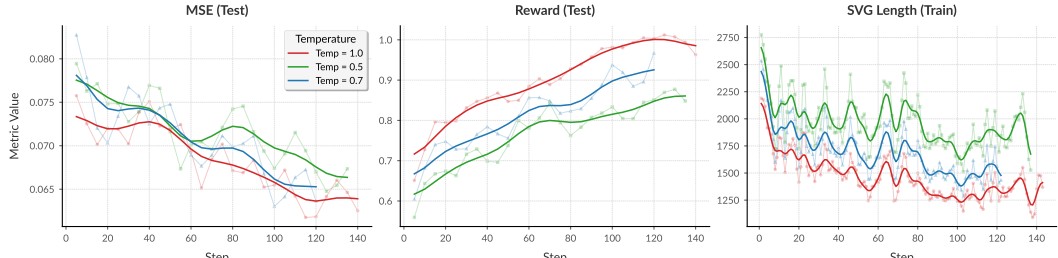

Figure 4: **Ablation on Sampling Temperature.** Keeping the sampling temperature high is critical for promoting roll-out diversity. Test MSE, Reward, and SVG Length measurements consistently improve. We find that increasing the temperature up to 1.2 improves exploration, but values beyond this lead to unstable behavior and diverged outputs.

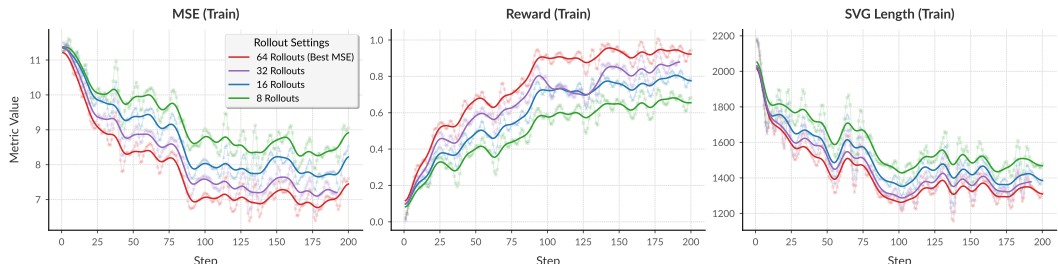

Figure 5: **Ablation on the Number of Roll-outs.** Increasing the number of roll-outs consistently improves MSE, reward, and SVG length. We report training curves, which offer clearer visibility by averaging over a large number of roll-outs.

which applies a quadratic penalty when the predicted length exceeds half the ground truth. This formulation allows moderate variation while discouraging overly long or redundant SVG sequences. We then apply clipping to constrain the reward within the interval $[-1, 1]$.

**Final Reward Aggregation** We integrate multiple reward signals by computing a weighted sum of individual components. Let $R_i$ be the $i$th reward function and $w_i$ its corresponding weight, for $i = 1, \ldots, K$. The final reward is given by $R_{\text{total}} = \sum_{i=1}^{K} w_i R_i$. This formulation allows fine-tuning the contribution of each component, ensuring a balanced and flexible training signal.

# 4 Experiments

## 4.1 Experimental Setup

The main paper focuses primarily on the Im2SVG experiments, as this setting offers a well-defined and visually grounded framework for evaluating the SVG performance gains achieved with RLRF. We also explore the Text2SVG setting, presented primarily in Appendix A.2, where RLRF demonstrates strong generalization beyond image-conditioned generation. All experiments are conducted using publicly available vision-language models (VLMs), specifically the Qwen2.5-VL [Bai et al., 2025] and Qwen3 [Yang et al., 2025a] families (the latter used only for Text2SVG). We also include the StarVector-1B model [Rodriguez et al., 2025b], which is trained specifically for the Im2SVG task, and a fixed input resolution of $224 \times 224$. In contrast, Qwen2.5-VL supports adaptive input resolutions and benefits from broad pretraining, enabling stronger general knowledge. However, Qwen models still exhibit limited performance on SVG-specific tasks without targeted fine-tuning.

**Supervised Fine-Tuning on SVGs (SVG-SFT)** We fine-tune Qwen2.5-VL models (3B and 7B) on the Im2SVG task using a cleaned subset of 1.7M image-SVG pairs from the SVG-Stack dataset [Rodriguez et al., 2025b], resulting in the SVG-SFT models. Training runs used $4 \times 8$ H100 GPUs (3B model) or $8 \times 8$ H100 GPUs (7B model) for $\sim 4$ days over 3 epochs, with learning rate $1e{-}5$, batch

size 1024, and context length 32k tokens. Although Qwen2.5-VL supports up to 128k tokens, we limit it to 32k to fit  90% of the data given memory constraints.

**Reinforcement Learning from Rendering Feedback (RLRF) on SVGs**   For the Im2SVG task, we further post-train the Qwen2.5-VL models, as well as StarVector-1B (which can be viewed as an SVG-SFT model), using RLRF. We begin by filtering the SVG-Stack dataset to select 20k high-entropy samples that are rich in visual detail and SVG complexity (each with over 500 tokens). Details of this data curation process are provided in Appendix B.2. During training, we use the GRPO algorithm with a rollout batch size of 32 images per step. For each image, 64 rollouts are generated, resulting in 2,048 rollouts per training step. We train for 500 steps in total, covering 16k unique images, significantly fewer than the 1.7M samples used in SVG-SFT. Training was completed in approximately 3 days using 4 nodes, each with 8×H100 GPUs. For Text2SVG, we use Qwen3-8B, a text-only model, and train it using image caption datasets (Flickr30k and MM-Icons), using only the captions as inputs (no SVG supervision). The model is prompted to `<think>` before generating SVG code. We train on a single node with 8×A100 GPUs for 4 days, using a rollout size of 16 and a batch size of 32 per step, for 1000 steps, corresponding to 16k unique captions. Across all RLRF experiments, we use a learning rate of $1e-5$ with 70% decay every 100 steps. KL regularization is disabled (KL coefficient $= 0$), with a clipping threshold $\epsilon = 0.4$ and sampling temperature set to 1.1.

**Exploration of Rendering-Based Rewards and Hyperparameters**   Starting from a Qwen2.5-VL-3B model fine-tuned with SVG-SFT, we apply RLRF. We use the smallest model to reduce exploration cost, while ensuring that the findings generalize to larger models. We run 800 steps (25.6k images) to study the impact of different rendering-based reward functions, and 150 steps (4.6k images) for hyperparameter ablations involving sampling temperature, KL divergence, and the number of rollouts per input.

## 4.2   Evaluation

**Baselines.** We compare our approach against image processing methods (Vtracer [Vision Cortex, 2023], Potrace [Selinger, Peter, 2024], PyAutoTrace [Weber, Martin, 2024]), deep learning methods (LIVE [Ma et al., 2022], DiffVG [Li et al., 2020]), and LLMs and VLMs including Claude3.7 Sonnet [Anthropic, 2024], GPT-4o [Hurst et al., 2024], Gemini Family [Georgiev et al., 2024] and open-source models like Qwen2.5VL [Bai et al., 2025] or Llama4 [Meta, 2025] families. These baselines span a wide range of modeling approaches, scales, and architectures.

**Benchmarks.** For Im2SVG, we report results on `SVG-Stack-Hard`, a curated subset of 500 visually complex and diverse SVGs selected from the original SVG-Stack (see Appendix B.2 for more details on datasets). We also evaluate on additional benchmarks including `SVG-Emoji`, `SVG-Fonts`, and `SVG-Icons` [Rodriguez et al., 2025b]. For Text2SVG evaluation, we use the `MM-Icon` and `MM-Illustration` [Yang et al., 2025b], as well as `Flickr30k Captions` [Young et al., 2014].

**Metrics.** For Im2SVG, we use *MSE* and *SSIM* for pixel-level fidelity, and *DINO Score* [Oquab et al., 2023] and *LPIPS* Zhang et al. [2018] for perceptual similarity. *Code Efficiency* is the negated mean difference between ground truth and predicted SVG token counts. Positive values indicate more compact outputs. Ideally, the score is near zero, reflecting compression without loss of fidelity. For Text2SVG, we evaluate with CLIP similarity and an LLM-based judge (Qwen2.5-VL-72B) for text–image alignment. Additional details on metrics are provided in Appendix B.1.

# 5   Results

## 5.1   Main Results

**Im2SVG Results.**   Table 1 presents the Im2SVG results. Image processing methods like LIVE and VTracer achieve strong reconstruction scores, especially in MSE, by densely fitting paths to the image. However, this leads to artifacts and extremely long SVGs, often exceeding 7k tokens and reaching up to 100k, as reflected in their low code efficiency. Closed-source VLMs perform well overall. While they do not achieve perfect reconstructions, their performance improves with scale and benefits from SVG-rich pertaining. Open-source VLMs lag behind. Smaller Qwen2.5VL models (3B and 7B) require heavy prompting and struggle to generate SVGs. Larger versions (32B and 72B) can produce SVGs, but still fail at accurate reconstruction.

Table 1: **RLRF Boosts SVG Generation Performance.** We compare baselines on the Im2SVG task using the `SVG-Stack-Hard` test set. Lower MSE/LPIPS and higher DINO/SSIM indicate better performance. Code Efficiency reflects token compactness, with values near zero being ideal. Open VLMs lag behind, while closed models perform well without SVG-specific tuning. Image processing methods achieve strong scores but generate verbose, inefficient code. LIVE scores highest but with high sampling cost. RLRF sets a new state-of-the-art with consistent gains across all metrics.

| Model | ↓ MSE | ↑ SSIM | ↑ DINO | ↓ LPIPS | Code Eff. | Time(s) |
|---|---|---|---|---|---|---|
| *VLMs (Open-Source)* | | | | | | |
| Qwen2.5VL-32B-Instruct | 23.62 | 55.46 | 82.38 | 35.83 | +1.3k | 58 |
| Qwen2.5VL-72B-Instruct | 23.20 | 55.72 | 81.68 | 34.14 | +1.4k | 62 |
| Llama4-Scout (109B) | 20.98 | 58.58 | 83.72 | 33.37 | +1.4k | 57 |
| Llama4-Maverick (400B) | 20.67 | 59.26 | 85.61 | 31.75 | +1.3k | 61 |
| *VLMs (Closed-Source)* | | | | | | |
| Gemini-Flash-1.5 | 20.38 | 59.65 | 84.70 | 33.27 | +1.2k | 59 |
| Gemini-Flash-2.0 | 19.31 | 60.21 | 86.53 | 32.10 | +1.1k | 63 |
| Gemini-1.5-Pro | 20.19 | 60.75 | 84.17 | 33.02 | +1.2k | 58 |
| Claude 3.7 Sonnet | 17.73 | 69.33 | 79.80 | 28.42 | +1.4k | 62 |
| GPT-4o-1120 | 16.92 | 66.91 | 89.00 | 27.55 | +1.3k | 60 |
| *Image Processing Methods* | | | | | | |
| Im2VEC | 18.10 | 76.50 | 69.20 | 29.10 | -4.3k | <1 |
| Potrace | 8.15 | 77.28 | 89.23 | 19.10 | -7.3k | 12 |
| DiffVG | 6.64 | 81.23 | 86.12 | 20.5 | -19.7k | 31 |
| PyAutoTrace | 4.71 | 87.44 | 95.68 | 10.71 | -99.7k | <1 |
| VTracer | 4.25 | 87.94 | 95.75 | 11.66 | -12.9k | <1 |
| SuperSVG | 3.05 | 83.30 | 82.70 | 13.50 | -65.6k | <1 |
| LIVE | 2.22 | 88.11 | 93.45 | 7.23 | -18.3k | 1,243 |
| *RLRF Results on SVG Base Models* | | | | | | |
| StarVector-1B-Base | 4.60 | 87.00 | 96.00 | 9.22 | -800 | 64 |
| **+RLRF** (ours) | **3.46** | **88.00** | **98.00** | **7.51** | **-127** | 23 |
| *Δ Improvement* | *-1.14* | *+1.0* | *+2.0* | *-1.71* | *+763* | *-41* |
| Qwen2.5VL-3B-Instruct | 23.31 | 62.28 | 69.26 | 35.30 | +1.5k | 24 |
| +SVG-SFT (ours) | 9.48 | 78.40 | 92.60 | 17.44 | -2.5k | 67 |
| **+RLRF** (ours) | **4.79** | **88.76** | **95.97** | **10.97** | **199** | 48 |
| *Δ Improvement* | *-4.69* | *+10.36* | *+3.37* | *-6.47* | *+2.7k* | *-19* |
| Qwen2.5-VL-7B-Instruct | 23.10 | 61.40 | 78.00 | 33.80 | +765 | 37 |
| +SVG-SFT (ours) | 8.60 | 79.40 | 93.00 | 16.58 | -2.8k | 73 |
| **+RLRF** (ours) | **1.03** | **95.10** | **98.70** | **3.08** | **-334** | 63 |
| *Δ Improvement* | *-7.57* | *+15.70* | *+5.70* | *-13.50* | *+2.5k* | *-10* |

**RLRF significantly boosts the SVG capabilities of StarVector-1B, Qwen2.5VL 3B and 7B**, improving both reconstruction accuracy and code efficiency. Qualitative examples in Figure 2 highlight the **notable gains achieved with RLRF**: it consistently generates coherent and well-aligned SVGs, while other methods often exhibit misalignments due to their lack of rendering awareness.

**Text2SVG Results.** Figure 3 presents qualitative samples from four models: Qwen3-7B with RLRF, Qwen3-32B, GPT-4o, and Gemini 1.5 Pro. **Using only 16k natural captions** from Flickr30k and MM-Icons, with no paired SVG supervision, and our text-image alignment rewards, **RLRF enables Qwen3-8B to generalize to the Text2SVG task, consistently producing SVGs that closely align with user prompts** (see Table 4 and Figures 3, 9). Qwen3-32B, despite its larger size, lacks the SVG generation capabilities and rendering awareness required for high-quality outputs. It often produces results that are misaligned or semantically incorrect. GPT-4o and Gemini 1.5 Pro generate more coherent SVGs, but frequently rely on oversimplified representations and struggle with spatial layout and fine-grained detail. While Qwen3-8B with RLRF achieves strong performance, it still faces

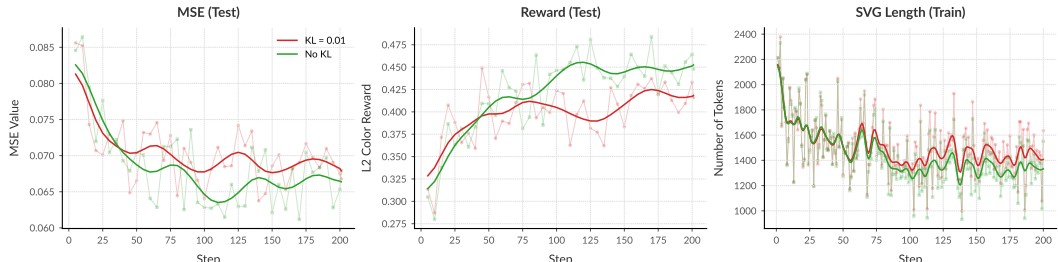

Figure 6: **Ablation of the KL Term.** Removing the KL term improves reward learning by avoiding early saturation. We attribute this to the conditional distribution $p(\text{SVG} \mid \text{image})$ already being well-regularized by the rewards. RLRF benefits from this added flexibility without instability.

Table 2: **Ablation on the importance of SFT for RLRF.** We evaluate the Qwen2.5VL-7B model under different training configurations to assess the role of supervised fine-tuning on the Im2SVG task (SVG-SFT). The baseline is the instruction-tuned model, which has limited SVG capabilities. Applying SVG-SFT provides a strong performance boost. Running RLRF directly on the instruction-tuned model yields poor results, while the best performance is achieved by combining SVG-SFT followed by RLRF.

| Method | MSE ↓ | SSIM ↑ | DINO ↑ | LPIPS ↓ | Code Eff. |
|---|---|---|---|---|---|
| Baseline (Instruct) | 23.10 | 61.40 | 78.00 | 33.80 | -765 |
| Instruct + SVG-SFT | 8.60 | 79.40 | 93.00 | 16.58 | +2,827 |
| Instruct + RLRF | 14.23 | 67.23 | 81.12 | 26.32 | -321 |
| Instruct + SVG-SFT + RLRF | **4.42** | **89.09** | **96.12** | **10.41** | **+173** |

challenges with precise Bezier curves and often relies on basic geometric primitives. These limitations could be addressed through further model scaling and more diverse training data. Additional results and analysis are provided in Appendix A.

**Models trained with RLRF demonstrate strong SVG generalization.** We stress-test these models on out-of-distribution benchmarks, including SVG-Emoji, SVG-Fonts, and SVG-Icons, and show in Table 7 that performance improves despite no exposure to these datasets during training. Qualitative examples, such as the "apple" and "strawberry" cases in Figures 1 and 2, further illustrate this effect. These samples include visual styles with shadows and layered effects that were never encountered during training and differ significantly from the training distribution. While the SVG-SFT baseline fails to produce coherent outputs under such shifts, RLRF enables the model to develop a deeper understanding of SVG structure, allowing it to generate well-aligned and reasonable outputs, even if not perfect. The generated SVGs are not only visually faithful to the input but also syntactically clean, editable (Figure 12), and practical for downstream applications. Additional results and discussion are provided in Appendix A.

## 5.2 Ablation Studies

**SVG Exploration Matters.** As shown in Figures 4 and 5, using a sampling temperature of 1.0 encourages diverse roll-outs, resulting in richer reward signals and more effective learning. Increasing the number of roll-outs per input, from 8 to 64, consistently improves performance. We observe no signs of reward saturation, suggesting that even more roll-outs could further enhance results. Figure 6 shows the effect of removing the KL divergence penalty in the GRPO objective. We find that **excluding the KL term improves reward progression, avoids early saturation, and reduces computation**, by eliminating the need to query the reference model. Our interpretation is that the conditional SVG distribution $p(x_s \mid x_c)$ is naturally narrow, thanks to the structured SVG syntax and strong reward signals. The KL term thus provides limited benefit and may unnecessarily constrain exploration. In addition, omitting the KL penalty improves computational efficiency by eliminating the need to evaluate the reference model during each training step.

**Running RLRF directly on instruction-tuned checkpoints fails**, as shown in Table 2. These models lack the SVG proficiency needed to select appropriate primitives and explore effectively. SVG-SFT is essential to build baseline SVG fluency, while RLRF sharpens these skills for reliable Im2SVG and Text2SVG generation.

**Ablation of Rewards.** The choice of reward has a significant impact on learning speed. Figure 13 shows that combining L2 and L2-Canny accelerates convergence compared to using either alone. Adding DreamSim further accelerates convergence. While L2-based rewards favor precise reconstruction (lower MSE), they lack semantic coherence, as reflected in lower DINO scores. The best performance is achieved by combining pixel-level and semantic rewards, along with a length penalty for code compactness. More detailed analysis is provided in Appendix A.4.

## 6 Conclusion

We have shown that reinforcement learning (RL) can be highly effective for inverse rendering tasks, where models generate code that is rendered into images. By designing tailored rewards, we significantly enhance SVG reconstruction and generation capabilities in vision-language models (VLMs). We introduced RLRF (Reinforcement Learning from Rendering Feedback), a novel approach for fine-tuning autoregressive VLMs on SVG generation using RL. RLRF equips models with a stronger understanding of the SVG code space, enabling them to produce more efficient, semantically aligned, and visually accurate outputs, even on benchmarks and SVG types not seen during training. We first apply supervised fine-tuning (SFT) to give models a strong foundation in SVG generation. RLRF then drives exploration through rendered rollouts, computes automatic rewards, and optimizes generation accordingly. This results in a fully automated, high-quality training signal powered by a composite reward that balances pixel-level accuracy, semantic alignment, and code efficiency. The results are compelling: models trained with RLRF consistently generate SVGs with deeper structural understanding, improved visual fidelity, and significantly better code efficiency.

**Broader Impact** While our focus is on SVGs, the core idea behind RLRF generalizes in principle to other inverse rendering code generation tasks. This includes HTML/CSS/JavaScript for web development, LaTeX/TikZ for scientific visualizations, 3D rendering code, and CAD modeling. We believe RLRF offers a general framework for improving structured, code-driven visual synthesis.

**Acknowledgements** We sincerely thank Ghazwa Darwiche, Christian Hudon, Fanny Rancourt, and Marie-Ève Marchand for their invaluable administrative and technical support. This work was supported by the Natural Sciences and Engineering Research Council of Canada and Mitacs. Chris Pal acknowledges the Canada CIFAR AI Chair. This work was partially supported by the MITACS program, which we gratefully acknowledge.

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

# Appendix

This appendix provides additional details and analyses for the RLRF approach. It is organized as follows:

- **Appendix A** presents extended quantitative and qualitative results, including visualizations of SVGs generated from both image and text inputs. These results demonstrate how RLRF outperforms supervised fine-tuning (SVG-SFT) and other strong baselines.

- **Appendix B** details the experimental setup, including datasets, evaluation metrics, and implementation specifics.

- **Appendix C** provides further elaboration on the RLRF approach, covering architectural choices, training objectives, implementation techniques, and limitations.

- **Appendix D** offers a comprehensive overview of related work in SVG generation and reinforcement learning for structured code output.

## Table of Contents

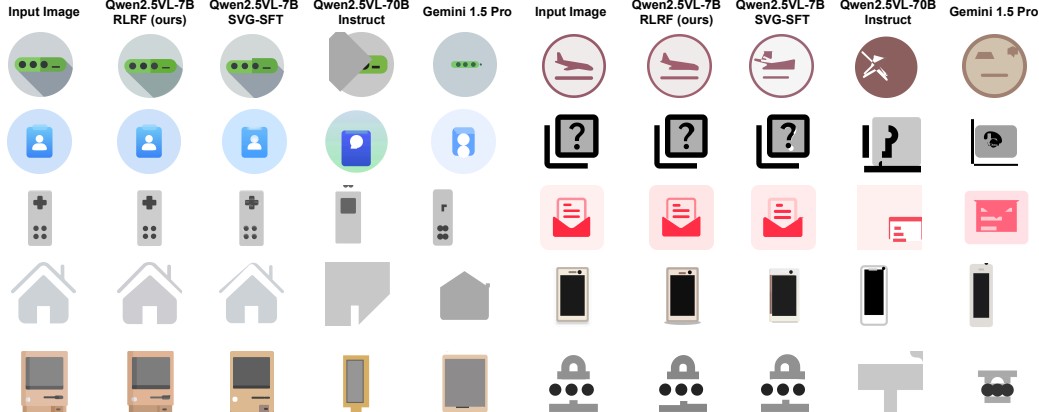

Figure 7: **Im2SVG Results Visualization.** We present qualitative comparisons of our RLRF outputs on test samples from the SVG-Stack-Hard benchmark, alongside generations from Qwen2.5VL-7B (finetuned on SVGs via SVG-SFT), Qwen2.5VL-70B (not specifically trained on SVGs but proficient at SVG code generation), and Gemini 1.5 Pro. Our RLRF significantly enhances the performance of Qwen2.5VL-7B, enabling it to produce SVGs that are more geometrically accurate, color-consistent, and visually aligned with the input images.

## A    Additional Experiments and Results

This section provides additional experiments and results. Specifically, we present visualizations of the generated SVGs for both the Im2SVG and Text2SVG tasks, along with comprehensive quantitative scores. We also evaluate the generalization ability of RLRF by analyzing its performance on out-of-distribution test sets. Finally, we include an ablation study to assess the impact of different reward configurations.

### A.1    Im2SVG Results

We show visualizations of results for **Im2SVG** in Figure 7, where we compare outputs from our RLRF with those from the SVG-SFT version of the Qwen2.5VL-7B model, as well as larger baselines including Qwen2.5VL-70B and Gemini 1.5 Pro. The examples are drawn from the challenging SVG-Stack-Hard test set. The SVG-SFT model often exhibits common failure modes such as misaligned shapes, incorrect spatial composition, or missing visual elements. In contrast, our RLRF significantly improves geometric accuracy and semantic alignment. For instance, in several samples requiring intricate spatial layouts or multiple object parts, RLRF consistently generates well-structured and complete SVGs, correcting the errors of the baseline models. Surprisingly, even compared to much larger models like Qwen2.5VL-70B and Gemini 1.5 Pro, our fine-tuned 7B model produces outputs that are more visually faithful and aligned with the input images. This highlights the impact of our reinforcement learning approach in improving structured image-to-code generation.

### A.2    Text2SVG Results

Table 4 presents quantitative results for the Text2SVG task across three benchmarks: Flickr30k (proposed here), MM-Icons, and MM-Illustrations [Yang et al., 2025b]. These datasets allow for broad comparison across a wide range of methods. For our RLRF models, Flickr30k and MM-Icons captions are used during training, meaning the prompts are in-distribution. However, the corresponding SVGs are not used, making the generation task itself zero-shot. MM-Illustrations serves as an out-of-distribution evaluation set.

We compare both open-source and closed-source VLMs, none of which were explicitly trained on these benchmarks. For baseline models, we report the best available results, including those from MM-Icons and MM-Illustrations as documented in [Yang et al., 2025b]. We are actively working to expand the set of evaluated models and report additional scores. We also introduce a new metric, Accurate, which is computed only on results for which ground-truth comparisons are

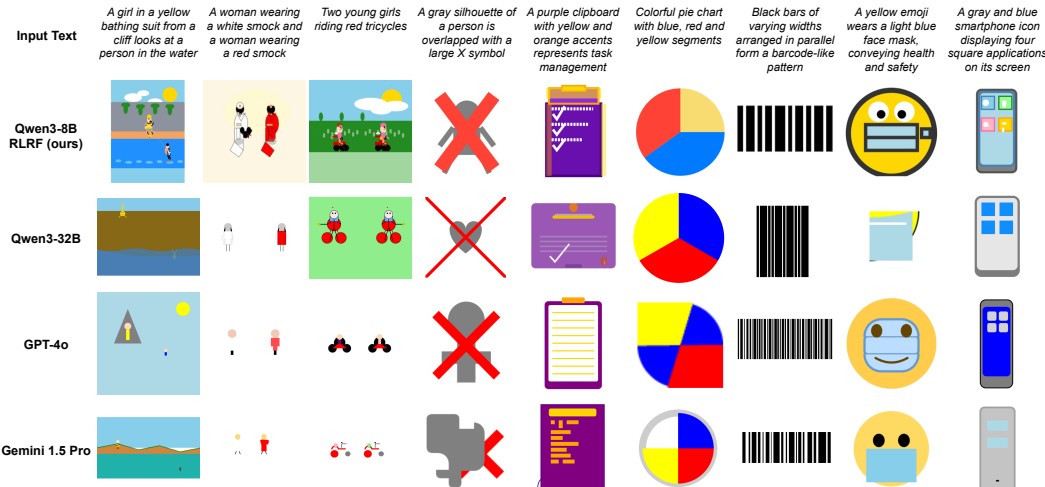

Figure 8: **Text2SVG Results Visualization.** We present qualitative comparisons of our RLRF outputs on test samples from the Text2SVG benchmark, alongside generations from Qwen3-32B, GPT-4o, and Gemini 1.5 Pro. Our method effectively aligns Qwen3-8B to produce high-quality, semantically rich, and visually coherent SVGs.

feasible. The results show that while RLRF does not yet outperform the OmniSVG baseline, it demonstrates strong generalization and SVG generation capabilities. Notably, our method approaches OmniSVG's performance despite not being trained on any paired SVGs from the target datasets. Among all models, Claude 3.7 achieves the strongest results, potentially due to specific tuning for SVG tasks. Nonetheless, RLRF showcases a promising path toward rendering-aware, reward-driven SVG generation.

We show Text2SVG visualizations in Figure 8, where we compare our method (Qwen3-8B RLRF) with larger baseline models, including Qwen3-32B, GPT-4o, and Gemini 1.5 Pro. Our approach consistently produces SVGs that are both semantically aligned with the input text and visually coherent. For example, in the "girl in a yellow bathing suit" prompt, our model captures the full scene with correct composition and multiple characters, whereas other models either omit key elements or fail to structure the layout meaningfully. Similarly, for more abstract prompts like the "purple clipboard with yellow and orange accents", our method generates a recognizable object with the correct semantic attributes, while others produce vague or incorrect outputs. Notably, models like GPT-4o and Gemini often misinterpret structural or compositional cues, especially in prompts requiring multiple interacting objects. Figure 9 presents two qualitative examples that illustrate the model's reasoning process after applying RLRF, as well as the alignment between the generated SVGs and the input prompts. While the outputs are not yet fully aesthetic, **it is remarkable that the model demonstrates such strong generalization despite being trained only with text captions from natural image datasets, i.e., without paired SVG supervision.** The reward signal is provided solely by a VLM (Qwen2.5VL-7B), which evaluates the rendered SVGs, as detailed in Appendix C.2. *Notably, the model has never seen an image during training (it does not have an image encoder), yet it learns to produce visually coherent graphics through this reward signal.* These results demonstrate that RLRF enables smaller models to outperform larger baselines in structured visual grounding tasks. We also tested RLRF on Qwen3-3B but found it ineffective, as the small size of the model makes it lack the SVG proficiency required to benefit from reinforcement learning.

## A.3  Generalization across SVG Benchmarks

We tested the generalization capabilities of our trained RLRF models and found that the reinforcement learning stage equips them with strong out-of-distribution robustness. RLRF models can handle new visual domains and vector styles that were never seen during training. This is shown in Table 3, presenting results on three out-of-distribution datasets (SVG-Emoji, SVG-Fonts, and SVG-Icons) which were entirely excluded from training. Rows marked "RLRF (ours)" correspond to models first instruction-tuned on SVG-Stack (1.7M samples), then refined with RLRF using a curated subset

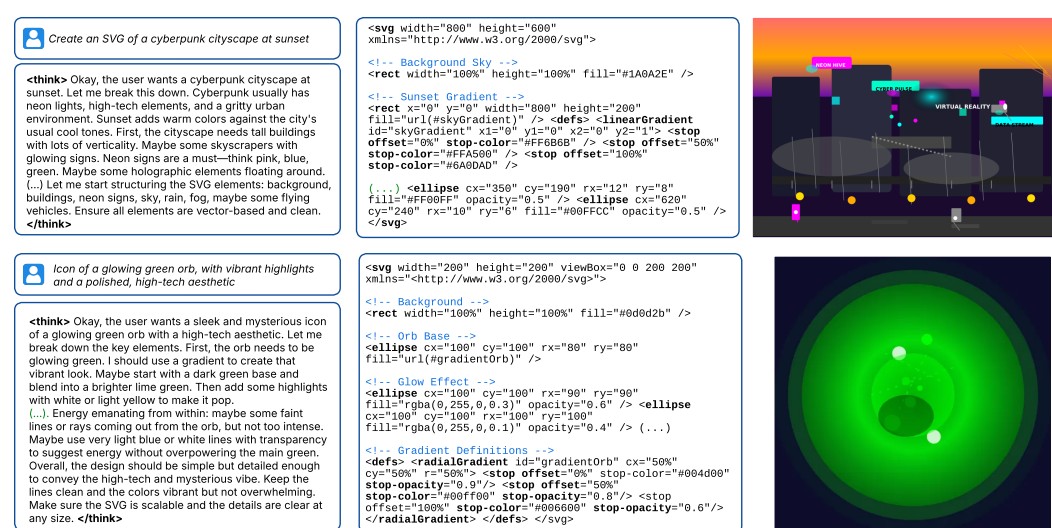

Figure 9: **Text2SVG Generation Examples from RLRF.** We show samples generated by the Qwen3-8B model after applying RLRF training. Each example begins with a reasoning-style planning phase, followed by the generation of SVG code. The two prompts—one describing a cyberpunk scene at sunset and another depicting a mysterious green orb—are rendered into long and complex SVGs that closely match the inputs. Notably, the model achieves this despite its relatively small size (8B), and such quality was not attainable before reinforcement learning. Larger models and more training data are expected to further improve results, as current trends show no signs of saturation.

of 16k images (SVG supervision is not used in this stage, only rendered rollouts are rewarded). Across all perceptual fidelity metrics (lower MSE and LPIPS, higher SSIM and DINO), **RLRF delivers significant improvements over its own supervised (SVG-SFT) baseline and over all other open-source or commercial VLMs.** For example, on *SVG-Emojis*, the Qwen2.5VL-7B SVG-SFT model improves from 6.39 MSE and 90.99 DINO to **4.93** MSE and **93.50** DINO, while reducing the average SVG length by approximately 1,500 tokens. A similar improvement is observed on *SVG-Fonts*, where the MSE drops from 7.1 to **4.73**, despite no fine-tuning on that domain. *These results demonstrate that reward signals derived from rendered images generalize effectively to unseen domains*, even without direct supervision on those categories during optimization. In contrast, models like StarVector require explicit fine-tuning on each target dataset to achieve comparable performance, as shown by their task-specific tuning on *SVG-Fonts* and *SVG-Icons*.

We further observe that RLRF consistently outperforms SVG-SFT across all metrics on both *SVG-Emojis* and *SVG-Fonts*. However, on *SVG-Icons*, RLRF only improves perceptual metrics such as DINO and LPIPS, while MSE and SSIM scores are slightly worse. This discrepancy is likely due to the nature of the dataset, which is heavily skewed toward sparse line drawings on white backgrounds. In such cases, small misalignments in thin black strokes can cause large changes in pixel-based metrics, making MSE and SSIM less reliable indicators of perceptual quality. Notably, traditional image processing methods perform quite well in general, as they are highly optimized for this style of imagery. However, this comes at the cost of producing extremely verbose SVGs with unnecessarily long token sequences, which lack semantic structure and often result in less efficient, non-sharp outputs and with artifacts [Rodriguez et al., 2025b].

We show qualitative visualizations in Figures 10 and 11, highlighting the generalization capabilities of RLRF. Despite never being trained on these datasets, models fine-tuned with RLRF produce SVGs that closely match the input in both shape and style, often achieving near-perfect reconstructions. SVG-SFT provides a reasonable baseline and captures basic structure, confirming the importance of the supervised pretraining stage. However, it frequently exhibits notable misalignments. For example, in the "disk" icon, RLRF preserves overlapping circular segments more accurately, whereas SVG-SFT distorts the occlusion geometry. In the "dress" example, RLRF leverages symmetry more effectively, producing a more coherent and balanced shape. Other models, including GPT-4o, Claude 3.7, and Gemini 1.5 Pro, perform inconsistently and often fail to reconstruct fine details or preserve shape semantics. This highlights the advantage of reinforcement learning from rendering feedback, which

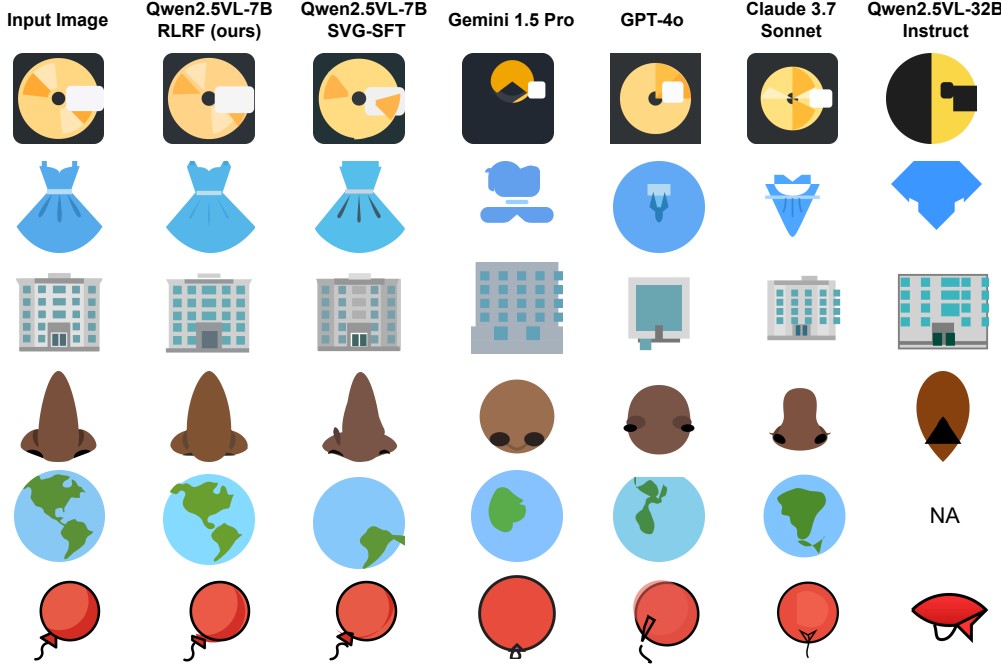

Figure 10: **Results on the SVG-Emoji test set.** The task is Im2SVG, where the first column shows the pixel-based input image and the remaining columns show generations from different models. This benchmark tests out-of-distribution generalization, as neither RLRF nor SVG-SFT models were trained on this dataset. RLRF shows clear improvements over SVG-SFT, with more coherent and visually accurate outputs. Other strong models like GPT-4o and Claude 3.7 also demonstrate notable generalization

enables RLRF to develop a stronger understanding of structure and style across out-of-distribution examples.

A limitation we observe on RLRF is the model's tendency to "give up"" on very complex inputs, producing code that diverges and then falls into loops. This likely stems from difficulty approximating intricate visual content. We believe this can be mitigated by scaling up the diversity and complexity of training data, especially images that require longer and more challenging SVG sequences.

**Image processing methods still excel at pixel error.** Vectorization pipelines like VTracer and LIVE achieve the lowest MSE values on inputs that are simple single-color glyphs with white backgrounds, where exact pixel reconstruction is relatively easy. However, these methods score lower on perceptual metrics like DINO, which emphasize sharpness and human-perceived fidelity. They also produce SVG code that is orders of magnitude longer and less efficient than any learning-based approach, lacking semantic alignment and practicality for editing or deployment.

**RLRF offers the best balance.** While image processing methods remain strong on raw pixel metrics, RLRF is the only approach that consistently improves pixel fidelity, structural similarity, semantic alignment, and code compactness. By combining reconstruction-based and efficiency-based rewards, RLRF learns to generate SVGs that are visually accurate, semantically meaningful, and compact, making it a strong candidate for real-world deployment where models must generalize to new domains and produce editable, efficient code.

### A.4 Ablation Study: Impact of Rewards

Figure 13 and Table 5 present an ablation study on the impact of different reward configurations introduced in Section 3.2. We begin with the Qwen2.5VL-3B model after completing the SVG-SFT stage. At this point, the model is already capable of generating valid SVGs that resemble the input

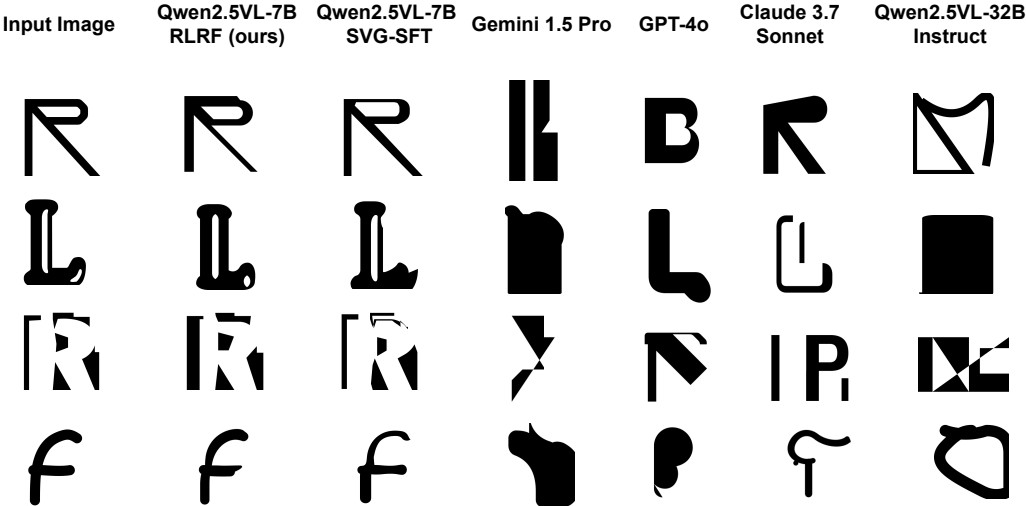

Figure 11: **Results on the SVG-Fonts test set.** The task is Im2SVG, with the first column showing the pixel-based input and subsequent columns displaying SVG generations from different models. This benchmark evaluates out-of-distribution generalization, as the models were not trained on font-like images. RLRF demonstrates strong vectorization capabilities, producing accurate and structurally clean SVGs. In contrast, the SVG-SFT model struggles with alignment and consistency, and all other models perform poorly on this dataset, often failing to capture basic shape.

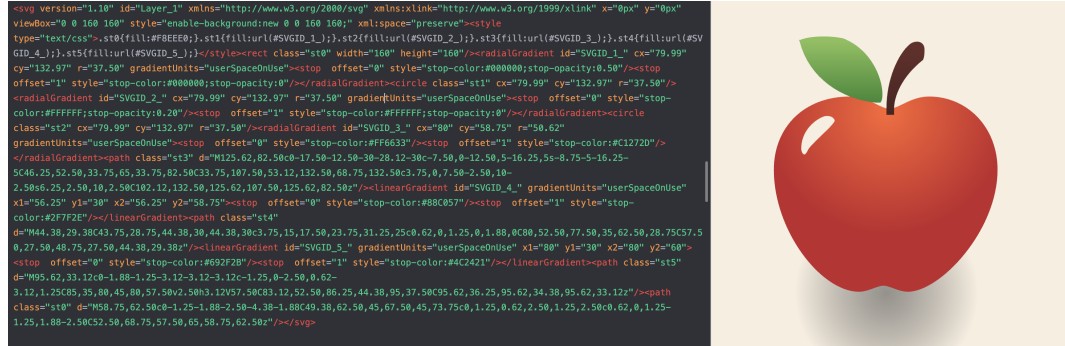

Figure 12: **Im2SVG Vectorization Result: Apple Example** This challenging sample highlights the effectiveness of RLRF. The image contains complex shapes, lighting effects, and semantic cues that require multiple SVG primitives: rectangles for the background, paths for the main structure, and gradients for shadows and highlights. Such examples are not present in the SVG-Stack dataset, which primarily contains simpler icons and logos. While SFT alone fails to reproduce this image, RLRF enables the model to generalize and successfully generate high-fidelity SVGs for out-of-distribution samples like this.

Table 3: **RLRF Generalization Results.** We test our RLRF models are evaluated on out-of-distribution datasets that were not used at any point during training. Despite not being exposed to these datasets, the models perform surprisingly well, demonstrating strong generalization capabilities. This is a notable improvement over previous SFT-based approaches, such as the StarVector-1B and 8B models, which only achieved good results when fine-tuned directly on each target dataset. Image processing methods also perform well in these settings, as they are specifically optimized for such domains. In contrast, other VLMs struggle to generalize beyond the SVG distributions seen during their pretraining, resulting in significantly lower performance on these out-of-distribution tasks.

| Model | ↓MSE | ↑SSIM | ↑DINO | ↓LPIPS | Code Eff. | ↓MSE | ↑SSIM | ↑DINO | ↓LPIPS | Code Eff. | ↓MSE | ↑SSIM | ↑DINO | ↓LPIPS | Code Eff. |
|---|---|---|---|---|---|---|---|---|---|---|---|---|---|---|---|
| | *SVG-Emoji* | | | | | *SVG-Fonts* | | | | | *SVG-Icons* | | | | |
| *VLMs (Open-Source)* | | | | | | | | | | | | | | | |
| Qwen2.5VL-32B-Instruct | 16.96 | 57.01 | 78.78 | 46.19 | -1850 | 30.44 | 61.40 | 78.68 | 25.76 | -1852 | 17.55 | 62.14 | 82.18 | 30.08 | -3864 |
| Qwen2.5VL-72B-Instruct | 17.94 | 58.25 | 74.38 | 45.83 | -2171 | 24.76 | 67.74 | 81.20 | 22.69 | -1906 | 24.50 | 60.97 | 79.50 | 26.55 | -3975 |
| Llama4-Maverick (400B) | 16.25 | 58.02 | 78.69 | 43.90 | -2106 | 22.82 | 67.35 | 82.81 | 22.11 | -1869 | 16.04 | 68.15 | 85.86 | 23.61 | -3968 |
| Llama4-Scout (109B) | 15.76 | 58.86 | 78.69 | 44.71 | -2188 | 23.75 | 65.85 | 80.25 | 23.25 | -1938 | 13.74 | 67.76 | 83.62 | 25.95 | -3948 |
| *VLMs (Closed-Source)* | | | | | | | | | | | | | | | |
| Gemini-Flash-1.5 | 15.42 | 59.47 | 80.31 | 44.57 | -1841 | 28.41 | 60.74 | 81.24 | 24.84 | -1823 | 18.29 | 63.10 | 83.93 | 27.86 | -3827 |
| Gemini-Flash-2.0 | 15.31 | 60.95 | 76.31 | 44.41 | -1866 | 23.31 | 65.98 | 83.88 | 23.11 | -1698 | 12.28 | 69.18 | 87.61 | 24.37 | -3662 |
| Gemini-1.5-Pro | 15.93 | 59.41 | 78.23 | 46.05 | -1842 | 27.19 | 62.92 | 81.11 | 24.11 | -1770 | 19.71 | 63.65 | 83.09 | 26.89 | -3815 |
| GPT-4o-1120 | 13.44 | 63.32 | 81.99 | 39.62 | -2122 | 20.73 | 69.13 | 86.39 | 21.22 | -1806 | 9.52 | 74.24 | 89.82 | 20.66 | -3884 |
| Claude 3.7 Sonnet | 11.43 | 64.95 | 89.10 | 35.86 | -1828 | 16.60 | 73.88 | 89.77 | 18.62 | -1695 | 7.83 | 76.79 | 93.30 | 17.57 | -3707 |
| *Image Processing Methods* | | | | | | | | | | | | | | | |
| Potrace | 6.70 | 78.00 | 88.20 | 26.70 | -9700 | 0.20 | 98.80 | 96.70 | 0.90 | -4200 | 0.40 | 97.30 | 97.20 | 2.30 | -12000 |
| VTracer | 0.80 | 89.40 | 98.10 | 7.40 | -15700 | 0.90 | 88.80 | 96.40 | 2.70 | -4500 | 1.70 | 91.40 | 94.00 | 6.20 | -20000 |
| PyAutoTrace | 1.10 | 90.20 | 97.50 | 7.70 | -94000 | 0.60 | 96.80 | 95.40 | 2.50 | -30800 | 1.40 | 93.70 | 94.60 | 5.30 | -56700 |
| DiffVG | 3.40 | 77.60 | 81.40 | 24.20 | -19700 | 0.70 | 95.90 | 82.10 | 5.10 | -19700 | 1.50 | 95.60 | 95.20 | 5.60 | -19800 |
| LIVE | 0.20 | 95.80 | 96.90 | 6.00 | -18300 | 0.10 | 97.70 | 95.60 | 1.30 | -18300 | 0.40 | 97.30 | 95.90 | 3.50 | -18200 |
| *RLRF Results on SVG Base Models* | | | | | | | | | | | | | | | |
| StarVector-1B | 6.30 | 82.00 | 92.90 | 21.70 | -4800 | 2.20 | 96.10 | 97.80 | 2.20 | -2400 | 2.60 | 93.10 | 97.50 | 4.00 | -3500 |
| StarVector-8B | 5.20 | 82.90 | 94.30 | 19.30 | -6700 | 2.90 | 94.60 | 98.20 | 3.00 | -3000 | 1.20 | 97.50 | 98.40 | 3.50 | -2800 |
| Qwen2.5VL-3B-Instruct | 20.25 | 59.84 | 68.84 | 46.32 | -2268 | 25.97 | 65.27 | 75.36 | 22.90 | -1939 | 21.43 | 65.60 | 77.41 | 24.53 | -4024 |
| +SVG-SFT (ours) | 6.74 | 81.38 | 89.34 | 25.78 | -2581 | 7.42 | 89.20 | 90.00 | 10.22 | -2060 | 5.56 | 89.07 | 85.94 | 12.68 | -4222 |
| **+RLRF** (ours) | 5.42 | 72.22 | 93.25 | 22.82 | -861 | 5.04 | 87.42 | 94.02 | 7.75 | -1574 | 6.13 | 82.59 | 91.20 | 11.64 | -3387 |
| Qwen2.5-VL-7B-Instruct | 18.07 | 60.32 | 70.93 | 45.75 | -2083 | 26.52 | 63.92 | 77.91 | 23.79 | -1888 | 13.42 | 69.41 | 80.59 | 25.21 | -3859 |
| +SVG-SFT (ours) | 6.39 | 81.85 | 90.99 | 24.12 | -2581 | 7.71 | 88.88 | 91.12 | 10.27 | -2060 | 6.16 | 88.34 | 85.22 | 13.37 | -4222 |
| **+RLRF** (ours) | 4.93 | 73.87 | 93.50 | 21.05 | -483 | 4.73 | 88.11 | 93.73 | 7.36 | -1550 | 7.38 | 81.64 | 90.04 | 12.05 | -3111 |

image in simple and moderately complex cases. However, it still lacks rendering awareness, which is necessary for achieving high pixel-level and semantic alignment.

We evaluate several reward configurations, including L2 loss alone, L2 Canny (which focuses on edges), DreamSim alone, DreamSim+Canny, and combinations that include the Length penalty. We also test the full composite configuration that integrates L2, DreamSim, Canny, and Length.

All experiments are evaluated on the SVG-Stack-Hard test set using four metrics: DINO Score, MSE, LPIPS, and SSIM.

Table 5 summarizes the best checkpoint scores across training steps. We observe that the best overall performance is achieved when using the full combination of rewards. This setup produces high semantic alignment (reflected by DINO Score), while maintaining strong reconstruction fidelity (low MSE and LPIPS, high SSIM).

The progression curves in Figure 13 highlight several trends:

- **DreamSim-based rewards alone saturate early** and yield weaker reconstruction scores, but show stronger perceptual alignment, as indicated by DINO Score. This suggests that DreamSim encourages high-level semantic consistency rather than precise pixel-level accuracy.

- **All configurations that include L2** lead to significantly better performance in MSE, LPIPS, and SSIM. L2+Canny slightly improves early convergence, while adding the Length term contributes to smoother and more compact SVGs.

- **The full reward combination consistently outperforms other setups**, offering both stable convergence and improved results across all metrics. This configuration balances pixel-level precision and perceptual alignment while also improving training stability.

Table 4: **Text2SVG Performance Across Diverse Datasets.** This table shows that RLRF improves Text2SVG performance over the base model (Qwen3-8B-Instruct before RL) across multiple datasets. Although the gains are less pronounced than in Im2SVG, this is partly due to the limitations of standard metrics like CLIP and Aesthetic, which are biased toward natural images and misaligned with SVG-like outputs (see Figure 3 for visual examples). Our proposed Accurate metric, which uses LLMs as judges, more clearly captures the improvements. We also report scores from OmniSVG [Yang et al., 2025b], the current state-of-the-art on MM-Icon and MM-Illustration. RLRF models consistently rank second, outperforming all other baselines.

| Model | Flickr30k | | | MM-Icon | | | MM-Illustration | | |
|---|---|---|---|---|---|---|---|---|---|
| | ↑ CLIP | ↑ Accurate | ↑ Aesthetic | ↑ CLIP | ↑ Accurate | ↑ Aesthetic | ↑ CLIP | ↑ Accurate | ↑ Aesthetic |
| *VLMs (Open-Source)* | | | | | | | | | |
| LLM4SVG-GPT2XL (3B) | 18.58 | 0.27 | 0.40 | 26.82 | 3.21 | 3.12 | 21.70 | 2.38 | 2.47 |
| Qwen2.5VL-32B-Instruct | 22.10 | 2.08 | 2.07 | 30.26 | 3.57 | 3.21 | 28.54 | 3.19 | 2.80 |
| Qwen2.5VL-72B-Instruct | 22.27 | 1.78 | 2.13 | 30.26 | 3.48 | 3.22 | 29.01 | 3.13 | 2.85 |
| Llama4-Scout (109B) | 21.94 | 1.92 | 2.42 | 30.31 | 3.56 | 3.27 | 29.00 | 3.30 | 2.99 |
| Llama4-Maverick (400B) | **23.26** | **2.36** | **2.51** | **31.17** | **4.01** | **3.54** | **30.18** | **3.76** | **3.25** |
| *VLMs (Closed-Source)* | | | | | | | | | |
| Gemini-Flash-1.5 | 22.09 | 1.59 | 2.16 | 30.28 | 3.37 | 3.23 | 29.70 | 3.16 | 3.05 |
| Gemini-1.5-Pro | 23.61 | 2.24 | 2.30 | 30.65 | 3.37 | 3.41 | 29.52 | 3.49 | 3.14 |
| Gemini-Flash-2.0 | 20.89 | 1.87 | 2.23 | 30.00 | 3.93 | 3.48 | 29.50 | 3.30 | 2.92 |
| GPT-4o-1120 | 25.00 | 2.43 | 2.48 | 31.92 | 4.10 | 3.57 | 31.53 | 3.92 | 3.32 |
| Claude-3.7-sonnet | **27.40** | **3.37** | **2.88** | **32.73** | **4.62** | **3.82** | **32.75** | **4.30** | **3.61** |
| *Text2SVG Models* | | | | | | | | | |
| Vectorfusion | - | - | - | 27.98 | - | - | 26.39 | - | |
| SVGDreamer | - | - | - | 29.23 | - | - | 27.95 | - | |
| Chat2SVG | - | - | - | 30.29 | - | - | 28.91 | - | |
| IconShop | - | - | - | 23.59 | - | - | 21.98 | - | |
| OmniSVG | - | - | - | **32.78** | - | - | **31.64** | - | |
| *RLRF Models* | | | | | | | | | |
| Qwen3-8B-Instruct | 22.50 | 2.74 | 2.53 | 30.09 | 3.77 | 3.35 | 29.05 | 3.63 | 3.15 |
| +**RLRF**(flickr) (ours) | **24.42** | **3.65** | **2.95** | 30.20 | 3.88 | 3.44 | 29.06 | **3.95** | **3.47** |
| +**RLRF**(icons) (ours) | 22.64 | 3.12 | 2.75 | **30.28** | **4.13** | **3.65** | **29.28** | 3.95 | 3.45 |

Table 5: **Ablation Study on Rewards.** Higher DINO and SSIM, and lower MSE and LPIPS indicate better reconstructions. This table shows the effect of different reward formulations. Using L2 achieves a strong MSE score, and L2+Canny further improves it. DreamSim and DreamSim-Edges alone result in poor performance, but their combination with Canny edges yields better outcomes. The best overall performance is achieved when combining all rewards using a weighted sum. We clearly observe a substantial improvement from the baseline (Qwen2.5VL-3B after SVG-SFT, before RL) to the final model trained with RLRF.

| Reward(s) | ↓ MSE | ↑ SSIM | ↑ DINO | ↓ LPIPS |
|---|---|---|---|---|
| Baseline (no RL) | 7.48 | 78.40 | 92.6 | 17.44 |
| L2 | 4.77 | 88.25 | 95.85 | 10.90 |
| L2 Canny | 4.60 | 88.30 | 95.60 | 10.95 |
| L2 + L2 Canny | 4.50 | 88.25 | 95.65 | 10.95 |
| DreamSim | 4.90 | 87.90 | 95.90 | 11.30 |
| DreamSim Canny | 4.85 | 87.90 | 96.00 | 11.35 |
| DreamSim Canny + L2 Canny + Length | 4.75 | 88.30 | 96.00 | 10.80 |
| All rewards (weighted sum) | **4.55** | **88.45** | **96.00** | **10.75** |

- L2 alone remains competitive for minimizing MSE, as expected for a pixel-focused reward, but lacks the broader alignment capabilities offered by perceptual and structural signals.

These findings confirm the effectiveness of combining multiple reward signals (pixel-based, semantic, and code efficiency-based), for optimizing SVG generation in RLRF.

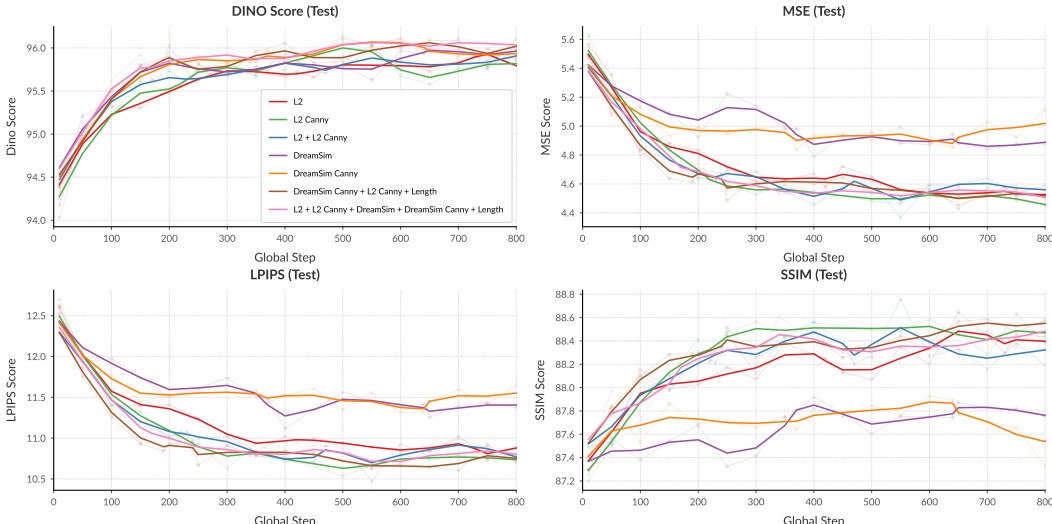

Figure 13: **Ablation on the Impact of Rewards.** We show the evolution of test metrics during RLRF training under different reward configurations. Using L2 alone leads to strong reconstruction performance, particularly in MSE. DreamSim on its own performs poorly, but combinations that include DreamSim alongside other signals tend to yield better results. The best overall performance is achieved by combining all reward components, which provides a stronger and more stable learning signal. This setup balances pixel-level accuracy (as reflected in MSE, LPIPS, and SSIM) with perceptual alignment (indicated by a higher DINO score).

## A.5 Cases of Reward Hacking

We observe several instances where the model learns to exploit the reward function without actually improving output quality, a behavior known as reward hacking. Below, we highlight one of the most notable cases and the corresponding mitigation strategy.

**Small ViewBox Hack.** The model learns to produce SVGs with extremely small viewboxes, for example: `<svg ... viewBox="0 0 1 1">`. This causes the renderer to generate an extremely low-resolution image, and the input image is similarly resized for comparison. As a result, most information is lost, and the image-based reward becomes artificially high.

This issue arises because we originally used the predicted SVG's viewbox to guide rendering resolution. To fix it, we enforce rendering using the reference image size and aspect ratio derived from the input image, ensuring a fair and stable reward computation.

**SVG Length Collapse.** In some cases, we observe that the model progressively generates shorter and shorter SVGs until it reaches a collapse point, after which generation diverges entirely and becomes unusable. This behavior is driven by the length deviation reward defined in Equation 6, which continues to incentivize shorter outputs up to half the ground truth length.

The issue arises because lengths below $\frac{1}{2}L_{\text{gt}}$ still receive increasingly higher rewards, peaking at 1 when exactly half the length is reached. However, this unintentionally encourages the model to keep shrinking the SVG below that threshold.

To address this, we explored assigning a fixed reward of $-1$ when the predicted length falls below half the ground truth. In practice, we found a more stable solution by reducing the weight of the length reward to 0.1 in early training, then gradually increasing it as RLRF progresses. Once the model has undergone sufficient RL training, it is no longer biased toward producing overly short sequences.

**Text-in-Image Hack.** In the Text2SVG setup, where the model receives a textual instruction and generates corresponding SVG code, we observed a common reward hacking behavior. The model learns to exploit the reward signal by using the `<text>` SVG primitive to render the exact prompt

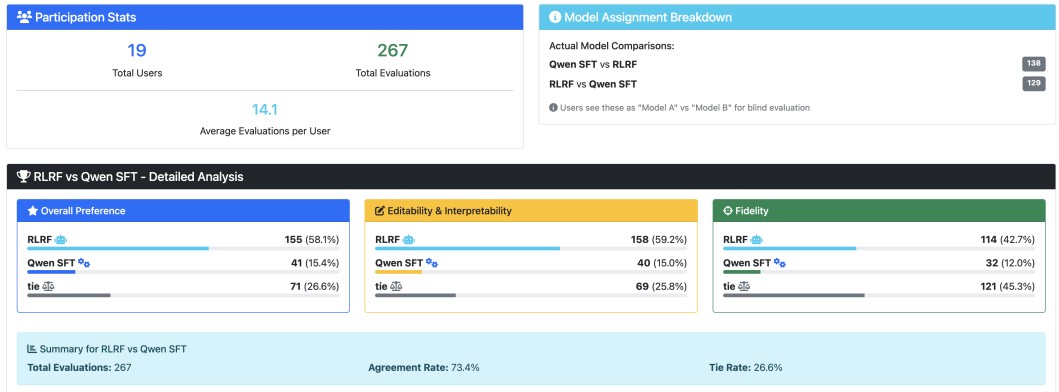

Figure 14: Statistics about the human evaluation study and main results, favouring RLRF-trained methods.

string directly onto the image. This artificially boosts similarity scores, especially when using CLIP-based rewards, since the rendered image containing the input text aligns closely with its textual embedding. Figure 9

To mitigate this, we preprocess the SVG before rendering and strip out any `<text>` elements or related primitives that visually encode the input prompt. We also strengthen reward robustness by incorporating a Qwen2.5-72B-based evaluator as part of the final reward mix, which helps reduce reliance on shallow visual-textual shortcuts.

## A.6 User Study on Editability and Fidelity

We ran a user study to evaluate the editability and fidelity of SVGs produced by four model variants: *StarVector*, *StarVector + RLRF*, *Qwen2.5VL-7B + SFT*, and *Qwen2.5VL-7B + RLRF*. We sampled 50 examples from the SVG-Stack-Hard dataset, covering a wide range of primitives and structural complexity.

For each example, participants were shown the ground truth image together with paired outputs before and after RLRF. They were asked to rate the following aspects:

- **Fidelity:** The extent to which the generated SVG matches the target image.

- **Editability:** Whether the SVG is clean, structured, and easy to modify, for example through appropriate use of primitives and absence of redundant paths or unused style attributes.

We collected a total of 267 evaluations from 18 participants, including three professional designers. All model outputs were anonymized, and the interface randomized the presentation order for each task.

In terms of *Fidelity*, RLRF was preferred in 42.7 percent of comparisons, SFT in 12.0 percent, and the remaining 45.3 percent were ties. Regarding *Editability*, RLRF was preferred in 59.2 percent of comparisons, SFT in 15.0 percent, and 25.8 percent were ties.

These results show that RLRF improves both fidelity and editability compared with SFT. Although SFT already achieves strong fidelity and ties with RLRF on many straightforward examples, RLRF performs better on the more challenging cases without reducing visual quality. Participants consistently favored the SVGs produced by RLRF in terms of structure and ease of modification. Designers specifically pointed out that SFT sometimes introduced unused attributes or redundant styles, while RLRF produced more compact and interpretable paths and a cleaner overall organization.

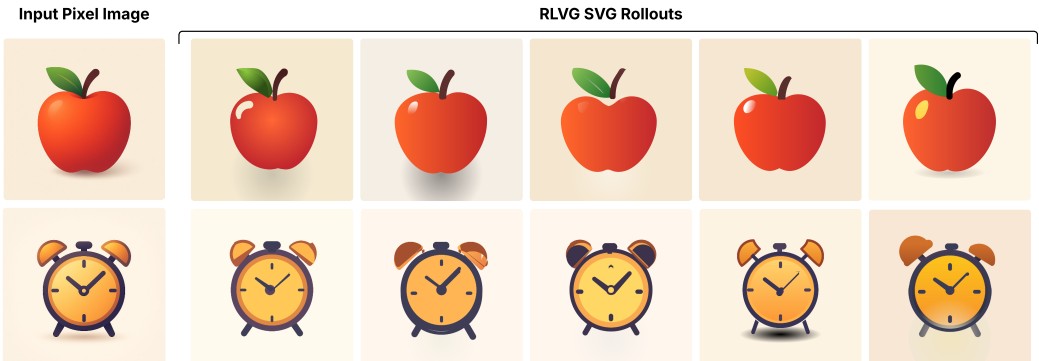

Figure 15: **RLRF Improves SVG Generation.** Given an input image (left), we show five SVG generations produced by the model after applying RLRF. By rendering its own predictions and receiving rewards for accuracy and compactness, the model improves over time.

# B  Experimental Setup

## B.1  Metrics

In addition to the metrics described in Section 4.2, we provide further details specific to the Text2SVG task. We first employ the CLIP Score, following prior work [Wu et al., 2023, Rodriguez et al., 2025b, Yang et al., 2025b], along with CLIP Aesthetics. These metrics capture general image-text alignment. However, we observed that they do not correlate well with the visual characteristics of the SVG images generated by our models when RLRF is applied to general-purpose instruction-tuned models such as Qwen2.5VL.

To address this, we introduce a dedicated Text2SVG Accuracy Score ("Accurate"), which uses a vision-language model (specifically, Qwen2.5VL-70B) as a judge. The VLM is prompted to rate the generation along several axes. The prompt used is shown in Prompt 1 and was carefully designed and tuned using a held-out validation set to ensure that model-based scores align with human preferences.

## B.2  Datasets

**Curating SVG Stack**  To train our model, we require large-scale data to capture the fundamental structures of SVG. For this, we leverage the SVG-Stack dataset [Rodriguez et al., 2025b], which consists of SVG files scraped from GitHub. We rasterize these files to obtain paired image-SVG examples.

The original SVG-Stack contains 2.1M samples. We preprocess this data by rounding decimals to two significant figures, removing XML headers, and filtering out samples with excessively long URLs or embedded base64 images, which could lead the model to memorize irrelevant content. After cleaning, we retain 1.7M high-quality examples, which we use to train the model in the SVG-SFT stage, where it learns to generate SVGs from images as faithfully as possible. The SVGs are rendered using CairoSVG [Kozea, 2023], and image augmentations follow the protocol introduced by LLaVA [Liu et al., 2023].

**SVG-Stack-Hard Test Set**  We construct the `SVG-Stack-Hard` benchmark to address several limitations of the original SVG-Stack evaluation set, which contains noisy, overly simple, and short samples. The full test set is also relatively large (3k samples), making evaluation slow, and includes some broken or empty (white) SVGs.

To improve quality and difficulty, we first filter out broken SVGs, white-background SVGs, and samples with low color entropy. We then retain only SVGs with at least 500 tokens to ensure complexity. Next, we cluster the remaining samples using DINO image features and perform stratified sampling to select 600 examples. Finally, we manually verify and curate this set to ensure it includes visually intricate and moderately challenging samples.

**Implementation Details**   We use the LLaMA-Factory codebase [Zheng et al., 2024] to conduct our supervised fine-tuning (SFT) experiments. For reinforcement learning, including our GRPO-based approach, we build on EasyR1 [Zheng et al., 2025] and VERL [Sheng et al., 2024]. We leverage vLLM [Kwon et al., 2023] for sampling during rollout generation, as it enables highly optimized decoding with high throughput and low latency. This is particularly important for SVG generation, which involves long context sequences.

**Training Details**   For GRPO, we use a clipping parameter $\epsilon = 0.4$. During RLRF training, we adopt the AdamW optimizer with `bf16` precision. The optimization settings include a maximum gradient norm of 1.0, a weight decay of $1 \times 10^{-2}$, and no learning rate warmup (`lr_warmup_ratio = 0.0`). We enable full parameter sharding via Fully Sharded Data Parallel (FSDP) full model sharding. We keep the vision tower frozen during SVG-SFT and RLRF

---

**Prompt 1:  Used for VLM as a Judge Score (Accurate)**

```
<|im_start|>user<|vision_start|><|image_pad|><|vision_end|> You are an
impartial evaluator of SVG/icon renderings.
--------- RUBRIC ----------
Alignment Score (0-5) - ''Does the image depict what the text describes?''
0 - Completely unrelated:  no shared objects, themes, or context.
1 - Very weak match:  one minor element overlaps, but overall scene/concept
is different.
2 - Weak match:  a few elements overlap, yet key objects or the main action
differ.
3 - Partial match:  primary objects/actions align, but notable details or
context differ.
4 - Strong match:  image reflects the description with only small,
non-critical discrepancies.
5 - Perfect match:  image fully and accurately depicts every essential
detail of the description.
Aesthetics Score (0-5) - ''Overall visual quality:  clarity of meaning +
first-impression appeal.''
0 - Unusable:  broken or illegible; no clear subject; chaotic or noisy.
1 - Very poor:  subject partly recognizable but ugly-obvious errors,
off-proportion shapes, harsh or clashing colors.
2 - Poor:  conveys the subject but feels rough; unbalanced layout, dull/flat
styling, sparse detail.
3 - Fair:  subject clear at first glance; proportions mostly correct;
acceptable composition and palette with minor flaws.
4 - Good:  rich detail, harmonious colors, balanced negative space; polished
with only subtle imperfections.
5 - Excellent:  instantly communicates its subject; perfect proportions and
composition; refined details, beautiful color harmony-production-ready.
--------- TASK ---------
Rate the image on **both** scales above.  Return **only** this JSON
object-nothing else:
'''json { "alignment_score":  <integer 0-5>, "alignment_reason":
"«=100-word justification>", "aesthetics_score":  <integer 0-5>,
"aesthetics_reason":  "«=100-word justification>" } Description:
<|im_end|><|im_start|>assistant
```

---

## C   RLRF Method

### C.1   Multimodal Architecture

To perform SVG generation using the autoregressive approach (see Figure 16), we adopt a vision-language model (VLM) [Alayrac et al., 2022, Liu et al., 2023] composed of a decoder-only language model [Brown et al., 2020] that generates SVG code tokens one at a time, and an image encoder that processes visual inputs. In tasks that do not require an image, the image encoder can be omitted, and the model reduces to a text-only LLM. For Im2SVG, the model receives an image as input and generates SVG code as output, treating the code as a plain text sequence in standard SVG format. In

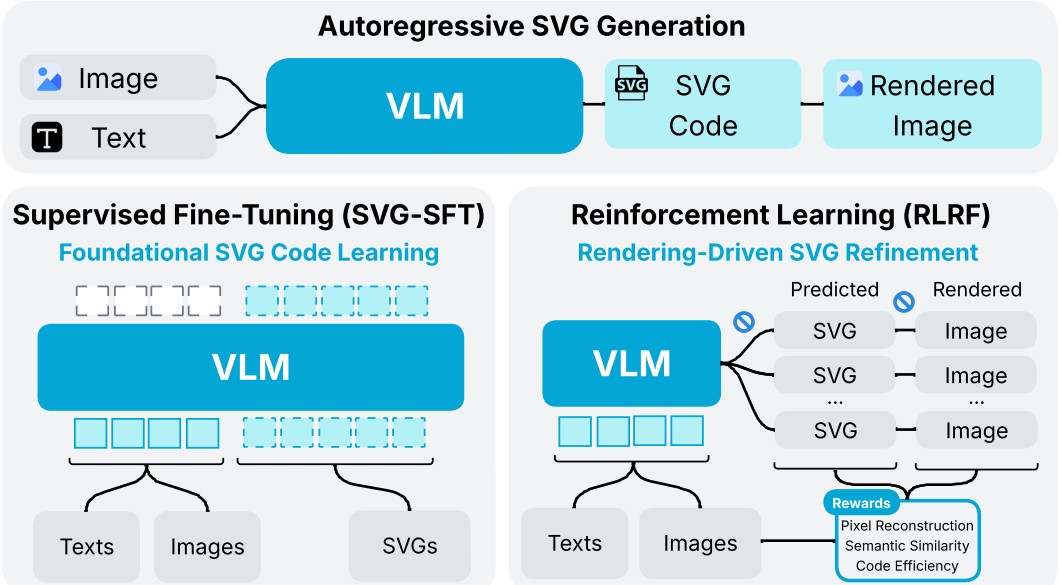

Figure 16: **Overview of Autoregressive SVG Generation and RLRF.** (Top) Autorregresive SVG generation pipeline. An image or a prompt is given to a VLM to produce SVG code, which is then rendered into an image. (Bottom-left) SVG Supervised Fine-Tuning. The VLM is trained with teacher forcing to generate the SVG code from the training data. (Bottom-right) Reinforcement Learning for Vector Graphics. Given the same input (image or text), the model generates several SVG roll-outs, which are then rendered, and a reward is calculated to update the model.

contrast, Text2SVG is a purely textual task, where a strong language model alone may be sufficient to generate valid SVGs from text prompts.

**Vision-Language Model Design**     VLMs combine an image encoder with a decoder-only language model to process and generate multimodal content. The image encoder, often a Vision Transformer (ViT) [Dosovitskiy et al., 2020] or a convolutional network [Krizhevsky et al., 2012], converts the input image into a sequence of high-dimensional vectors called visual tokens. These are not discrete tokens like in language, but continuous embeddings representing image regions.

To make these embeddings compatible with the language model, they are passed through linear projection layers [Liu et al., 2023]. This allows the language model to attend to the visual context during generation. Since large images can produce many visual tokens, some models add modules to reduce their number and retain only the most relevant ones. For example, Q-Former [Li et al., 2023a] uses learnable queries to extract a smaller set of tokens, while Perceiver Resampler [Alayrac et al., 2022] and AlignVLM [Masry et al., 2025] use resampling and sparse attention to compress the visual information efficiently.

Once the visual tokens are projected, they are combined with any textual prompt and fed into a decoder-only language model. This model generates output tokens one at a time, using both the image and text context. In our case, the output is SVG code. The model learns to translate visual inputs into structured sequences of SVG instructions that can be rendered to reconstruct the original image. This setup allows the model to reason about both the appearance and structure of visual content and generate precise vector representations.

**Model Choice: Qwen2.5-VL**     We perform our experiments using the Qwen2.5-VL [Bai et al., 2025] family of models, which are general-purpose VLMs designed for diverse multimodal tasks. Qwen2.5-VL uses a dynamic-resolution vision transformer (ViT) as the image encoder and a high-capacity language model decoder based on the Qwen2.5 series. This combination allows it to scale to long contexts, adapt to varied image resolutions, and perform complex vision-language reasoning. The vision tokens extracted by the ViT are prepended to the text tokens and jointly processed in the decoder.

**Comparison to StarVector** Compared to StarVector [Rodriguez et al., 2025b], which is specifically designed for SVG generation, Qwen2.5-VL is a more generalist model not specialized for code or vector formats. StarVector employs a CLIP-based image encoder and a code-generation-optimized decoder (e.g., StarCoder) with an adapter to bridge modalities. It is trained directly on SVG-Stack, a large dataset of real-world SVGs, making it more attuned to the syntax, structure, and compositional semantics of SVG code. In contrast, Qwen2.5-VL benefits from broader multimodal training but lacks SVG-specific pretraining, which we address through supervised fine-tuning and reinforcement learning.

**Design Tradeoffs** The generalist nature of Qwen2.5-VL enables strong transfer learning capabilities and flexibility across tasks but makes it initially less precise at SVG generation. Our method, RLRF, fine-tunes Qwen2.5-VL to align its output with visual fidelity and code compactness through supervised fine-tuning (SVG-SFT) reinforcement learning (RLRF). This specialization helps close the gap with task-specific models like StarVector while maintaining broader instruction-following abilities inherent to Qwen2.5-VL.

**Context Length** Context length plays a crucial role in autoregressive SVG generation. Complex SVGs with fine-grained structure and intricate visual elements often require sequences that span tens or even hundreds of thousands of tokens. The Qwen2.5-VL model supports a context length of up to 128k tokens, which allows it to handle long and highly detailed SVG code sequences.

With the growing availability of models that support very long contexts, including recent work pushing towards hundreds of thousands or even millions of tokens [Liu et al., 2025, Meta, 2025], we expect context length to become less of a limitation in the near future. In principle, this trend makes large-scale vector representation increasingly feasible.

In our experiments, we cap the context length at 32k tokens to fit within GPU memory constraints. This is sufficient for all benchmark samples we evaluate, including a wide range of complex SVGs. In comparison, the StarVector model was limited to an 8k token context, which restricts its ability to model large or deeply nested SVG structures.

## C.2 Reward Details

**Text2SVG Rewards.** We incorporate image-text alignment signals to better guide the model toward generating SVGs that are consistent with the input text. However, CLIP-based rewards (including CLIP Score and CLIP Aesthetics) were not effective in this setting. These models provide weak supervision because the generated SVGs fall far outside the distribution of natural images on which CLIP was trained. As shown in Figure 8, the SVG outputs tend to follow an abstract visual style composed of primitives such as rectangles, spirals, and curves, which makes them poorly suited for CLIP-based evaluation.

To address this, we propose using a **VLM as a judge** reward. This approach relies on a strong vision-language model (VLM) to evaluate the quality of the generated SVGs. Specifically, we prompt Qwen2.5VL-7B, a smaller variant selected for its memory efficiency during rollout scoring, to assess whether the generated SVG accurately reflects the input prompt. This VLM-based reward aligns more closely with human preferences and provides a significantly stronger training signal compared to CLIP. The reward prompts are described in Prompts 2 and 3, and include axes for evaluating semantic accuracy, visual resemblance, and aesthetic quality.

During RLRF rollouts with Qwen3-8B, the model is trained to produce `<think>` traces before answering. We take advantage of this behavior by prompting the model to first plan the SVG content, and then generate the code. The resulting SVGs include inline comments for better structure and readability. Examples of this reasoning process are shown in Figure 9.

```
Prompt 2:  Used for VLM as a Judge Reward for Text2SVG (Easy)

<|im_start|>user<|vision_start|><|image_pad|><|vision_end|>Does the drawing
resemble the description:  "" [Yes/No]<|im_end|><|im_start|>assistant
```

## C.3   Implementation Strategies for Stable and Robust RLRF Training

We describe practical strategies that enabled stable and efficient training of our RLRF method.

**Dynamic Max Length.**   After the SVG-SFT stage, models often lack strong SVG code completion skills and can struggle with out-of-distribution or complex samples. This occasionally leads to failure cases where the model enters indefinite loops, repeating SVG commands until the maximum token limit is reached. These long and unproductive rollouts slow down RL training, as each rollout must be fully sampled and passed through multiple models.

To mitigate this, we implement a dynamic maximum length schedule. For each batch, we estimate the required output length using the ground truth SVGs and set the maximum length to the longest sample plus a small threshold $t$. This strategy significantly reduces rollout overhead early in training, encourages the model to generate shorter and cleaner sequences, and naturally fades in importance as training progresses and generation improves.

During inference, input images are resized using bilinear interpolation such that the shortest side is $512$ pixels, preserving the original aspect ratio. We sample with temperature and top-p equal to 0.5 and 0.9 respectively, using the best-of-n strategy: we generate five candidates ($n = 5$) and select the one with the lowest mean-squared error (MSE) relative to the target image.

## C.4   Limitations

As with prior work, our method is constrained by the context length of current models. While this remains a challenge, we note that recent models with extended context windows are beginning to address it. A more specific limitation of our RLRF experiments is the tendency for the model to become increasingly specialized in SVG generation, which may diminish its general instruction-following capabilities. Although straightforward solutions exist, we leave their exploration to future work.

Another key limitation is the inefficiency of GRPO training, which is bottlenecked by rollout generation. This process introduces significant GPU idle time during the RL stage[2]. Mitigating this overhead is an important direction for future optimization.

# D   Extended Related Work

## D.1   SVG Generation

SVG generation methods are typically categorized into three main approaches: classical image processing, latent variable models, and large language model (LLM)-based techniques. Traditional methods such as VTracer [Vision Cortex, 2023], Potrace [Selinger, Peter, 2024], and Autotrace [Weber, Martin, 2024] convert raster images into vector graphics by tracing contours and clustering regions. While effective for extracting shapes, these approaches produce long and unstructured SVGs composed of raw path commands. The resulting code is verbose, difficult to edit, and lacks higher-level semantic organization.

Deep learning has also been applied to this domain, particularly through differentiable rendering techniques that overcome the inherent non-differentiability of SVG rendering. DiffVG [Li et al., 2020] introduced a differentiable rasterizer, enabling the development of latent variable models [Carlier et al., 2020, Cao et al., 2023, Wang and Lian, 2021, Ma et al., 2022] that learn deep representations of SVG commands and the parameters of geometric primitives. These models use architectures based

---

[2]https://huggingface.co/blog/ServiceNow/pipelinerl

on variational autoencoders [Kingma and Welling, 2013], diffusion processes [Ho et al., 2020], or autoregressive decoders [Cao et al., 2023].

Compared to traditional methods, latent variable models offer more structured control and support tasks like interpolation and style transfer. However, they often rely on simplified subsets of the SVG syntax, producing outputs that are less compact and less interpretable. Additionally, these models are typically trained on narrow, domain-specific datasets, limiting their generalization to diverse SVG types, an area where classical image processing methods remain more broadly applicable.

More recent approaches frame SVG generation as a code generation task using large language models (LLMs). We refer to this as the Autoregressive SVG Generation approach. By tokenizing SVG code and generating it directly, LLM-based models bypass intermediate representations and learn to model the full SVG syntax end-to-end.

StarVector [Rodriguez et al., 2025b] was among the first to explore this direction, combining a code-centric LLM (StarCoder [Li et al., 2023b]) with a CLIP-based image encoder [Radford et al., 2021]. It achieved strong performance on large-scale image-to-SVG benchmarks. Follow-up works such as Beyond Pixels [Zhang et al., 2023], SVGEditBench [Nishina and Matsui, 2024, 2025], and related studies [Cai et al., 2023] explored editing, reasoning, and structured SVG generation. More recently, OmniSVG [Yang et al., 2025b] extended this line of work by leveraging the Qwen2.5-VL foundation [Bai et al., 2025].

Despite their progress, these models face a key limitation: they do not observe or evaluate the visual output of the SVG code they generate. As a result, they often produce SVGs that are syntactically correct but visually inaccurate or inefficient. Moreover, unlike latent models that can rely on differentiable rasterization, autoregressive VLM approaches operate over discrete token spaces, making both rendering and sampling non-differentiable.

Our method, RLRF, addresses this gap by introducing a reinforcement learning-based post-training strategy that incorporates visual feedback. By closing the loop between code generation and rendered output, we overcome the limitations of purely supervised fine-tuning. This reinforcement learning approach enables optimization of non-differentiable rendering quality using automatic reward signals. As a result, models trained with RLRF generalize more effectively across SVG generation tasks compared to standard supervised fine-tuning.

### D.2 Vision-Language Models (VLMs)

Early VLMs such as Flamingo [Alayrac et al., 2022], BLIP-2 [Li et al., 2023a], LLaVA [Liu et al., 2023], and GPT-4V [OpenAI, 2023] introduced a powerful paradigm by adapting frozen vision encoders and connecting them to pretrained unimodal LLMs. This is typically done through learned projection layers that map visual features into token-like embeddings. These models follow an autoregressive generation strategy [Vaswani et al., 2017], enabling them to process both images and text within a unified token stream and perform instruction-following and multi-turn conversations grounded in visual inputs.

In parallel, the use of LLMs for code generation has grown rapidly recently. Code can be seen as a separate modality, with structure and syntax that differ significantly from natural language. Progress in this area has been driven by the availability of large-scale code corpora [Kocetkov et al., 2022, Penedo et al., 2024] and the development of specialized coding models such as Codex [Chen et al., 2021], CodeGen [Nijkamp et al., 2022], and StarCoder [Li et al., 2023b].

Recent frontier models like GPT-4o [Hurst et al., 2024], Gemini [Georgiev et al., 2024], Claude [Anthropic, 2024], and Qwen2.5-VL [Bai et al., 2025] have expanded these capabilities further by incorporating larger and more diverse code datasets during pretraining, substantially improving performance on structured code tasks.

This convergence of multimodal models [Liu et al., 2023] and their use on structured output generation [Masry et al., 2025, Zhang et al., 2025, Feizi et al., 2025, Nayak et al., 2025] and code generation [Li et al., 2023b] enables a new class of problems: *inverse rendering code generation*, also known as *visual-code generation*). In these tasks, the model generates code that compiles or renders into visual content. Examples include SVG, TikZ, HTML, and CAD, which are used for generating graphics, diagrams, and illustrations. Prior efforts in this area include diagram and layout generation [Rodriguez et al., 2023b,a, Belouadi et al., 2023, 2024, Rodriguez et al., 2025a,

Belouadi et al., 2025, Nayak et al., 2025, Awal et al., 2025, Bechard et al., 2025] and text-to-CAD synthesis [Wang et al., 2025].

Although current models can learn valid code distributions and produce visually plausible outputs, they often suffer from hallucinations, limited long-range coherence, and poor generalization. A key limitation is the lack of feedback from the rendered environment. These models are never shown how their outputs look. Our work addresses this gap by enabling models to learn from their own rendered predictions, providing a visual feedback signal that improves both precision and generalization in code-driven visual tasks.

### D.3 Reinforcement Learning Post-Training

Reinforcement Learning (RL) has become essential for post-training fine-tuning of large language models (LLMs). Reinforcement Learning from Human Feedback (RLHF), using Proximal Policy Optimization (PPO) [Schulman et al., 2017], has become the standard for aligning LLM outputs with human preferences, improving tasks like summarization and instruction-following [Ziegler et al., 2019, Stiennon et al., 2020, Ouyang et al., 2022]. An alternative approach, Group Relative Policy Optimization (GRPO) [Shao et al., 2024, Guo et al., 2025], stabilizes training by normalizing rewards across batches, removing the need for a separate value network and reducing variance. In the visual generation space, GRPO has been successfully used for image editing [Ahmadi et al., 2025]. In code generation, frameworks like CodeRL [Le et al., 2022] employ an actor-critic approach where the critic predicts functional correctness to guide the actor [Le et al., 2022]. PPOCoder [Shojaee et al., 2023a] integrates PPO with execution feedback, using compiler results as rewards to fine-tune code generation models [Shojaee et al., 2023b]. StepCoder [Dou et al., 2024a] introduces a curriculum of code completion subtasks, optimizing exploration by masking unexecuted code segments [Dou et al., 2024b]. Additionally, Reinforcement Learning from Execution Feedback (RLEF) enables models to refine code iteratively based on execution outcomes, enhancing performance on complex tasks [Gehring et al., 2024]. In multimodal domains, approaches like ViCT use a visual critic to align generated UI code with input screenshots, improving fidelity in UI-to-code generation [Soselia et al., 2023]. RL has also been applied in architectural design, where agents learn to generate space layouts by optimizing spatial and functional constraints [Kakooee and Dillenburger, 2024]. Together, these methods demonstrate the power of RL in enhancing alignment, correctness, and efficiency across generative models.

While existing RL-based post-training methods focus primarily on functional correctness, they overlook visual quality. In contrast, RLRF introduces rendering feedback through a composite reward that jointly optimizes visual fidelity, semantic alignment, and code efficiency.

