# OpenReview forum: "Rendering-Aware Reinforcement Learning for Vector Graphics Generation"
_NeurIPS.cc/2025/Conference — NeurIPS 2025 poster_

### Official Review · Reviewer_CVwa · 2025-06-28

**Clarity:** 3
**Significance:** 3
**Originality:** 3
**Rating:** 4
**Confidence:** 2

**Summary:**

This paper presents an RL-driven method called RLVG, a two-stage approach for high-fidelity SVG code generation using pre-trained vision–language models (VLMs). For rewarding the model, the paper proposes sound loss functions, namely (1) pixel-level reconstruction (L2 and L2+Canny), (2) semantic similarity (DreamSim or CLIP), and (3) code efficiency (token-length penalty). Experiments on the SVG-Stack-Hard benchmark show RLVG markedly improves MSE, SSIM, DINO, LPIPS, and dramatically reduces SVG verbosity.

**Questions:**

Given the heavy computing reported, what strategies could reduce this cost (e.g., fewer roll-outs, model distillation)?
The results of the reconstructions diverge very differently (i.e., Fig 2), did the authors consider a small-scale user study to validate that RL-optimized SVGs align with human judgments of fidelity and editability?
The paper argues in future work (broader impact) that RLVG extends to other inverse-rendering tasks (HTML/CSS, TikZ). Do the authors have preliminary results or insights on such domains?

At the moment, I tend towards acceptance but I might change my mind based on the authors' and other reviewers' replies.

**Ethical Concerns:**

["NO or VERY MINOR ethics concerns only"]

**Final Justification:**

All raised concerns were addressed accordingly: I am happy to keep my rating as final decision! I hope the additional results, especially the user study, finds its way to the final manuscript!

**Limitations:**

Yes

**Paper Formatting Concerns:**

All good.

**Quality:**

3

**Strengths And Weaknesses:**

Strengths:
- The paper shows effort in evaluating their method. In more detail, it includes multiple metrics like pixel (MSE, SSIM), perceptual (DINO, LPIPS), and code-efficiency across multiple baselines and ablations.
- Extensive supplementary material answering core questions
- Introduces the first rendering-aware RL framework for autoregressive SVG generation, filling the gap left by non-differentiable rendering.

Weaknesses:
- RL stage requires thousands of roll-outs (e.g., 16K samples over 5 days on 32 H100s).
- Briefly mention the reward weight selection would be more appropriate in the main paper (instead of suppl. material), but that's just a style thing.

---

> ### Author Rebuttal · Authors · 2025-07-31
>
> Thank you for taking the time to review our paper. We sincerely appreciate your recognition of RLVG as the first rendering-aware reinforcement learning framework for SVG generation, as well as your positive remarks on the soundness of our reward design and the strength of our experimental evaluation. We're also glad to hear that the supplementary material helped clarify key aspects of the method.
>
> We have carefully considered all of your comments and suggestions, and we address each point in detail below.
>
> -------------
>
> ## **Q1. The RL stage requires heavy compute (e.g., 16K rollouts over 5 days on 32 H100s). Can this cost be reduced?**
>
> We agree that RL training is computationally demanding, a challenge also noted in prior works [1, 2]. This is primarily due to the memory and system-level requirements:
>
> - Each rollout step must load the **policy model**, **frozen reference model**, and **inference model**, all in parallel, which only fits on large-memory GPUs.
>
> - **During rollout collection, training GPUs remain idle**, and **during training, the inference GPUs sit idle**; this alternating workload limits throughput unless scaled to many GPUs. Using more GPUs increases training speed roughly linearly.
>
> The directions proposed by the reviewer are promising to reduce the cost:
>
> - **Model distillation**, as explored in DeepSeek-VL, is a compelling strategy: RL can be performed on a large model (which benefits more from feedback and scale), and the improved policy can then be distilled into a smaller, faster model for deployment.
>
> - **Reducing the number of rollouts** is effective, but with tradeoffs, as shown in Figure 6, performance improves steadily with more rollouts.
>
> ### **Experiment on cost reduction**:
>
> **We ran a new set of experiments using our latest optimized setup**, performing an ablation across four reward variants as suggested by Reviewer Lhdn. To r*educe computational cost*, we scaled down **from 16k to 1k samples**, reduced rollouts per sample **from 64 to 32**, halved the batch size, and cut GPU usage from **32 to 16 H100s**. Despite these constraints, the model showed *remarkable improvements*, confirming that **RLVG is effective even under limited resources and scales well with increased compute** (more rollouts, longer training).
>
> | Method       | Code Eff. | DinoScore | L2     | LPIPS  | SSIM   |
> |--------------|-----------|-----------|--------|--------|--------|
> | **SFT**      | -2.8k     | 93.00     | 8.60   | 16.58  |  79.40 |
> | **L2 Canny** | -334      | 98.70     | 1.03   | 3.08   | 95.08  |
> | L2           | -159      | 98.82     | 1.10   | 3.17   | 94.76  |
> | SSIM         | -210      | 98.64     | 1.30   | 3.34   | 94.76  |
> | L1           | -254      | 98.69     | 1.57   | 4.06   | 93.62  |
>
>
> A key optimization we plan to explore is introducing slightly *off-policy updates*, where training continues using cached rollouts from earlier policy versions, reducing idle GPU time. Techniques such as PipelineRL [1] support this without major stability loss.
>
> Finally, although expensive, **RLVG is highly data-efficient**. Compared to supervised fine-tuning (SFT) on Qwen2.5VL-7B, which required 2M samples over 14 days on 64 GPUs, our RL setup achieves stronger generalization using only ~16k rollouts over 5 days on 32 GPUs. Furthermore, we show in this rebuttal really good results with 1k rollouts and 16 GPUs.
>
> We believe that as the community advances RL infrastructure, these costs will continue to decrease, making rendering-aware RL more accessible in future work.
>
> [1]  ServiceNow & Hugging Face Research. (2025, April/May). PipelineRL. Hugging Face
> [2] DeepSeek‑AI, Guo, D., Yang, D., Zhang, H., Song, J., Xu, R., Zhu, Q., … Shao, Z. (2025, January 22). DeepSeek‑R1: Incentivizing reasoning capability in large language models via reinforcement learning. arXiv.
>
> -----------------
>
> ## **Q2. Could reward weight selection be included in the main paper rather than the supplementary?**
>
> Yes, we agree that this is an important detail that deserves to be in the main paper. We have now included the reward weight configuration directly in **Section 5.1**, with the following sentence (it will appear in the camera-ready version):
> > *“We find that the best-performing configuration uses weights:*
> > *\(w_{\text{L2}} = 1.0\), \(w_{\text{L2-Canny}} = 0.5\), \(w_{\text{DreamSim}} = 0.3\), \(w_{\text{LPIPS}} = 0.3\), and \(w_{\text{Length}} = 0.01\).”*
>
>
> -----------
>
> ## **Q3. Given the divergence in reconstructions (e.g., Fig. 2), did you consider a user study to validate fidelity and editability from a human perspective?**
>
>
> Thank you for the suggestion. **We conducted a user study** to assess **editability** and **fidelity** of generated SVGs across four model variants: StarVector, StarVector + RLVG, Qwen2.5VL-7B + SFT, and Qwen2.5VL-7B + RLVG.
>
> We sampled 50 examples from the SVG-Stack-Hard dataset, ensuring diversity across primitive types and structural complexity. For each sample, **participants were shown the ground-truth image and side-by-side outputs (pre- and post-RLVG), and asked to evaluate:**
>
> - **Fidelity**: How well the SVG matches the image
>
> - **Editability**: Whether the SVG is clean, structured, and easy to manipulate (e.g., use of primitives like <circle> or <text>, absence of redundant paths or styles)
>
> We collected **267 evaluations** from **18 participants**, including **3 professional designers**. Model outputs were anonymized and randomized in presentation.
>
> ### **Results**
> **Fidelity**: RLVG preferred in **42.7%** of cases, SFT in **12%**, tie in **45.3%**
>
> **Editability**: RLVG preferred in **59.2%**, SFT in **15%**, tie in **25.8%**
>
> The results demonstrate a **clear advantage for RLVG over SFT across both fidelity and editability**. While SFT already performs well in fidelity (tying with RLVG in nearly half the cases), RLVG matches this baseline and is **strongly preferred in the more challenging examples**. In terms of **editability**, participants **overwhelmingly favored RLVG**, citing *cleaner structure*, *better use of primitives*, and *more compact, interpretable code*. Importantly, these gains in efficiency come without any loss in visual quality, confirming that RLVG produces SVGs that are both accurate and significantly easier to edit.
>
> **Designers noted** that SFT often added unused styles or invalid attributes, while RLVG produced cleaner code with shorter, more editable paths
>
> ---------------
>
> ## **Q4. You mention broader applicability to other code-rendering tasks like HTML/CSS or TikZ. Do you have any preliminary results or insights in those domains?**
>
> Yes, **we have conducted early exploratory experiments** applying the **RLVG framework to HTML/CSS generation**. In these experiments, we used the **Qwen2.5VL-32B** model to generate HTML/CSS from web page screenshots, using the **WebMMU** dataset [1]. We treated the HTML/CSS outputs as rollouts, rendered them into webpages, and computed rewards based on the visual similarity between the rendered page and the input screenshot.
>
> We observed that **L2 and LPIPS-based rewards** are still useful in this setting, while edge-aware metrics like **Canny-L2** are less effective due to the structured and hierarchical nature of web layouts. Vision-language models (VLMs) also show strong potential as judges in this context.
>
> We hypothesise that **task-specific reward functions** will be crucial in this domain, ones that incorporate layout accuracy, DOM hierarchy, and code-level structure beyond appearance alone.
>
> In parallel, **we are encouraged to see that RLVG has already inspired follow-up work**. For example, **DeTikzify-v2.5-8B** [2] recently applied similar RL-based techniques to **TikZ code generation**, with strong results.
>
> These early findings support our belief that RLVG has the potential to significantly advance rendering-aware code generation across a wide range of visual programming domains.
>
> [1] Awal, Rabiul, et al. "WebMMU: A benchmark for multimodal multilingual website understanding and code generation." ICLR 2025 Third Workshop on Deep Learning for Code. 2025.
>
> [2] Belouadi, Jonas, Simone Ponzetto, and Steffen Eger. "Detikzify: Synthesizing graphics programs for scientific figures and sketches with tikz." Advances in Neural Information Processing Systems 37 (2024): 85074-85108.
>
> Thanks again for reviewing our paper.

---

> > ### Comment · Reviewer_CVwa · 2025-08-03
> >
> > I thank the authors for their very detailed answers and I am very happy to keep my tendency towards acceptance as final rating! Please insert the new results to the final manuscript!

---

### Official Review · Reviewer_STcx · 2025-06-29

**Clarity:** 3
**Significance:** 2
**Originality:** 2
**Rating:** 4
**Confidence:** 5

**Summary:**

This work proposes a new method for SVG generation. The core idea is to leverage reinforcement learning (RL) to improve both the visual quality (especially for complex semantics) and the code efficiency of generated SVGs. Additionally, the authors introduce a set of reward functions to guide the SVG generation process. Extensive experiments are conducted to demonstrate the method’s effectiveness, along with some in-depth analysis.

**Questions:**

Before evaluating code efficiency, is there any preprocessing applied to the SVG codes, such as removing redundant <g> tags or other forms of code cleaning, to ensure compactness? If so, please provide detailed information on the specific preprocessing steps used.

**Ethical Concerns:**

["NO or VERY MINOR ethics concerns only"]

**Final Justification:**

The authors’ response has addressed my concerns well. Accordingly, I would like to raise my rating.

**Limitations:**

Yes.

**Quality:**

3

**Strengths And Weaknesses:**

*Strengths*:

1. The paper is clearly written and provides insightful analysis (primarily in the supplementary materials).

2. The model demonstrates impressive performance on the image-to-SVG task, particularly in terms of generalizability to unseen cases (Fig. 1).

3. The experimental evaluation is extensive, effectively demonstrating the proposed method's advantages.

*Weaknesses*:

1. **Limited Novelty**: The proposed method appears to be an extension of StarVector (Rodriguez, 2025b) through the incorporation of RL. The method consists of two stages: SVG-SFT and RLVG. The first stage involves straightforward fine-tuning of a VLM using SVG data, while the main novelty lies in the design of reward functions for SVG generation. When StarVector is used as the base model, the method essentially becomes a post-training step. Furthermore, the performance improvements of StarVector+RLVG over StarVector across various metrics are relatively moderate.

2. **Lack of Qualitative Results**: The authors claim that incorporating RL improves visual fidelity and code efficiency. Therefore, more qualitative results are expected to support these claims. While quantitative results indicate improvements due to RLVG, it remains unclear how these translate into visual quality enhancements. In Fig. 2 (and Fig. 7 in the supplementary), results using Qwen2.5VL-7B as the base model are compared. It would be beneficial to provide side-by-side qualitative comparisons of StarVector and StarVector+RLVG on the same examples. Additionally, while RLVG seems to significantly improve code efficiency, the meaning of the reported numbers is unclear. For instance, using a ground-truth as reference, what does it look like when code efficiency is +1.3k, -18.3k, -800, or -127? Providing visual and code examples would help clarify these points.

3. **Evaluation of Code Efficiency**: The evaluation of code efficiency could be further improved. In my opinion, code efficiency should reflect the compactness and editability of the SVG code. A user study involving professional designers to assess the editability of the generated SVGs would provide a more comprehensive evaluation from an end-user perspective.

4. **Missing References and Baselines**:

(1) Several relevant image vectorization papers are not cited or compared against, including:
- [1] SuperSVG: Superpixel-based Scalable Vector Graphics Synthesis, CVPR 2024.
- [2] Vector Graphics Generation via Mutually Impulsed Dual-domain Diffusion, CVPR 2024.
- [3] Towards High-fidelity Artistic Image Vectorization via Texture-Encapsulated Shape Parameterization, CVPR 2024.

(2) When reporting performance on the text-to-SVG task, LLM4SVG (Xing, 2024) is missing as a baseline for comparison.

---

> ### Author Rebuttal · Authors · 2025-07-31
>
> Thank you for the time invested in reviewing our paper. We’re glad you found the writing clear, appreciated our insightful analysis, and recognized the strong generalization and performance of our model, as well as the overall quality of the experimental evaluation.
>
> We have carefully considered all of your comments, and we address each point in detail below.
>
> -----------------
>
> ## **Q1. Limited novelty vs. StarVector**
>
> We respectfully disagree with the characterization of our contribution as limited in novelty. While our method builds on StarVector as a base model, **we introduce a fundamentally new approach: applying reinforcement learning to visual code generation** tasks using autoregressive VLMs, **a new setting** where the generated code is rendered into an image, and the feedback is derived from that visual output.
>
> This is, to our knowledge, **the first method to successfully apply RL in the SVG VLM space**, addressing a range of challenges that are unique to the setting:
>
>  - **Sampling rollouts of SVG code** from a **large vision-language model** and computing **rendering-aware rewards** to **overcome the non-differentiability**.
>  - **Designing a reward suite for SVG and visual code generation** that balances pixel‑level fidelity, perceptual alignment, and code efficiency.
>  - **Overcoming optimization instabilities** tied to long‑sequence code generation and sparse reward signals.
>
> We believe **this work opens the door to a new class of rendering-aware fine-tuning techniques for visual code generation**, not just for SVG, but also for visual tasks. RLVG is a *foundational step* in a relatively unexplored but high-impact space.
>
> ---------------
>
> ## **Q1.1. Improvements of StarVector+RLVG over StarVector across various metrics are relatively moderate.**
>
> We would like to provide some context. While the improvement margin of StarVector+RLVG over the original StarVector is indeed smaller than what we observe with Qwen2.5VL+RLVG, **this is primarily due to limited optimization and resource investment on the StarVector branch**.
>
> Qwen2.5VL offered a more modular architecture and faster iteration cycles. StarVector is based on an earlier architecture that proved more difficult to tune, and as such, **for StarVector, we devoted fewer resources to its RL optimization**. Specifically, the **StarVector+RLVG experiments were conducted with 500 training samples and 8 rollouts per sample** and did not benefit from the full reward suite or extended training time.
>
> ### **That said, the observed improvements are still meaningful.**
>
> Across all metrics, StarVector+RLVG shows consistent gains of +1 to +2 points.** The drop in **code token count (Code Eff.) is dramatic**, highlighting real efficiency gains in the generated SVGs.
>
> **To strengthen this part of the paper, we have launched a new training run of StarVector + RLVG using our most optimized setup.**
>
> ----------
>
> ## **Q2. Lack of Qualitative Results. Can you provide more qualitative comparisons (e.g., StarVector vs. RLVG)?**
>
> We agree that the qualitative analysis in the submission (Figures 1, 2, 3, 8, 9) was primarily focused on Qwen2.5VL-based experiments and other baselines.
>
> **To address this, we have created two new figures comparing StarVector and StarVector+RLVG**, highlighting both visual outputs and code-level differences. The main takeaway is that **RLVG significantly improves layer alignment and structure** (as observed in the “strawberry” example of Figure 2), especially in complex SVGs. Additionally, it **reduces failure cases where the base model would previously hallucinate long, invalid SVG sequences or enter repetitive loops**. Unfortunately, we can not share images here, but they will be available in the camera-ready version.
>
> Additionally, we have added new qualitative figures from other benchmarks and ablations, covering image-to-SVG and text-to-SVG tasks.
>
> ---------------
>
> ## **Q2.1. Clarify what the code efficiency numbers mean with visual/code examples.**
>
> We appreciate the opportunity to clarify this further. **The code efficiency metric reports the token length difference between generated SVGs and ground-truth SVGs on the test set.**
>
> **Interpretation**:
>
> - *A value near 0 is ideal*: it indicates that the model produces SVGs similar in size to the reference.**
> - *A value much greater than 0 (e.g., +1.3k) often indicates that the model under-generates*, typically collapsing to overly simplified SVGs with missing details. This behavior is common in base VLMs without SVG-specific tuning, as shown in Table 1.
> - *A value far below 0 (e.g., –18k) means the model over-generates*, often producing excessively long path commands (as seen in image-processing-based methods) or falling into repetitive loops (as observed in some SVG-SFT-trained VLMs).
>
> *Our pipeline, after applying RL, consistently brings this value closer to 0*, by improving both **semantic precision** and **structural regularity** in the generated code.
>
> **We have added a new qualitative figure** that visually compares SVGs before and after RL fine-tuning, highlighting both the visual render and code length. These side-by-side examples illustrate how RL reduces redundancy while preserving fidelity.
>
> --------------------
> ## **Q3. Evaluation of code efficiency could be improved. Could you assess compactness and editability via a user study?**
>
> Thank you for the suggestion. **We conducted a user study** to assess **editability** and **fidelity** of generated SVGs across four model variants: StarVector, StarVector + RLVG, Qwen2.5VL-7B + SFT, and Qwen2.5VL-7B + RLVG.
>
> We sampled 50 examples from the SVG-Stack-Hard dataset, ensuring diversity across primitive types and structural complexity. For each sample, **participants were shown the ground-truth image and side-by-side outputs (pre- and post-RLVG), and asked to evaluate:**
>
> - **Fidelity**: How well the SVG matches the image
>
> - **Editability**: Whether the SVG is clean, structured, and easy to manipulate (e.g., use of primitives like <circle> or <text>, absence of redundant paths or styles)
>
> We collected **267 evaluations** from **18 participants**, including **3 professional designers**. Model outputs were anonymized and randomized in presentation.
>
> ### **Results**
> **Fidelity**: RLVG preferred in **42.7%** of cases, SFT in **12%**, tie in **45.3%**
>
> **Editability**: RLVG preferred in **59.2%**, SFT in **15%**, tie in **25.8%**
>
> The results demonstrate a **clear advantage for RLVG over SFT across both fidelity and editability**. While SFT already performs well in fidelity (tying with RLVG in nearly half the cases), RLVG matches this baseline and is **strongly preferred in the more challenging examples**. In terms of **editability**, participants **overwhelmingly favored RLVG**, citing *cleaner structure*, *better use of primitives*, and *more compact, interpretable code*. Importantly, these gains in efficiency come without any loss in visual quality, confirming that RLVG produces SVGs that are both accurate and significantly easier to edit.
>
> **Designers noted** that SFT often added unused styles or invalid attributes, while RLVG produced cleaner code with shorter, more editable paths
>
> ------------------
>
> ## **Q4. Missing References and Baselines**
>
> Thank you for pointing out these recent works. **We have updated the manuscript to include all four references.**
> We have run and reported results for SuperSVG [1] on the image-to-SVG task and LLM4SVG [4] on the text-to-SVG task. The results are now included in Tables 1 and 4.
>
> **Image2SVG results** on the svg-stack-hard test set
>
> | Model           | MSE ↓ | SSIM ↑ | LPIPS ↓ | DINO ↓ | Code Eff. |
> |-----------------|-------|--------|---------|--------|----------------|
> | LIVE            | 2.22  | 88.11  | 93.45   | 7.23   | -18.3k         |
> | **SuperSVG**    | 3.05  | 83.3   | 13.5    | 82.7   | -65.6k         |
> | **StarVector**  | 3.46  | 88.0   | 98.0    | 7.51   | -127           |
> | **RLVG (ours)** | **1.03**  | **95.1**   | **3.08**    | **98.7**   | **-334**           |
>
> **SuperSVG** achieves competent visual scores like MSE of 3.05, beating StarVector, but we observe that it **relies exclusively on long path sequences**, fitting curves densely without structure. This results in SVGs with **high visual noise and poor editability**, making them difficult to use in real-world design. **Our latest RLVG model shows the best scores**
>
> **Text2SVG results** on the Flickr30k, MM-Icon, and MM-Illustration benchmarks. For each model, we compute the average across these datasets to obtain aggregate summary scores.
>
> | Model                  | CLIP   | Accurate | Aesthetic |
> |------------------------|--------|----------|-----------|
> | Claude-3.7-sonnet      | 30.96  | 4.10     | 3.44      |
> | GPT-4o-1120            | 29.48  | 3.48     | 3.12      |
> | **LLM4SVG-GPT2XL** (3B)    | 22.37  | 1.95     | 1.99      |
> | **RLRF (OURS)**          | 27.89  | 3.83     | 3.29      |
>
> **LLM4SVG** is able to perform text-to-SVG generation and produces valid SVGs most of the time. However, it often *fails to follow the prompts in our benchmarks*, likely due to distributional shifts from its training data. As a result, it shows limited generalization capabilities.
>
> For the remaining two baselines,  [2] and [3], we were unable to run them due to the lack of publicly available code. However, we have added a brief discussion in Section 2 (Related Work):
>
> -------------
>
> ## **Q.5. Is there any processing applied to the SVGs before evaluating code efficiency?**
>
> **We do not apply any preprocessing aimed at compacting the SVG code**, such as removing redundant tags or cleaning object structures. The only processing step is a minimal repair routine: if an SVG is invalid (i.e., cannot be parsed by the `svgstr2paths(svg_string)` function from `svgpathtools`), we attempt to fix it by closing open paths. No other modifications or clean-up steps are applied.
>
> Thanks again for your review.

---

> > ### Comment · Reviewer_STcx · 2025-08-04
> >
> > Thank you for the authors’ response, which has addressed my concerns well. Accordingly, I would like to raise my rating.

---

### Official Review · Reviewer_RqsR · 2025-07-02

**Clarity:** 3
**Significance:** 3
**Originality:** 4
**Rating:** 5
**Confidence:** 4

**Summary:**

This paper proposes a reinforcement learning-based Scalable Vector Graphics (SVG) generation model named RLVG. By designing rendering output-aware rewards, based on image reconstruction, semantic similarity, and code efficiency, it fine-tunes pre-trained autoregressive VLMs to generate more precise, high-quality SVGs. Experiments demonstrate that RLVG achieves new state-of-the-art results on multiple benchmarks.

**Questions:**

1. The term "inverse rendering code generation problem" is not commonly used and should be revised or rephrased (e.g., avoid "known as"). "Perceptual inputs" should also be replaced with a more specific description (e.g., the caption in Figure 1 uses "text or image input," which is clear and concise).
2. Citation format: The current author-year style differs from the numeric format seen in other papers. Switching to numeric citations would save space in the main text, allowing inclusion of Appendix C.2, which contains crucial methodological details that should not be relegated solely to the supplementary material.
3. Issues with Figure 1:
(a) The caption states "compared to the input (image, SVG, or text)," which may confuse readers who recall this caption mentioning only "text or image input." Clarify the SVG input reference or align the caption with the introduction.
(b) While the stop sign is explained in the caption, adding a legend directly in the figure would improve clarity.
(c) The caption describes the example as "out-of-distribution with no associated ground truth SVG," conflicting with Lines 180–186, which require ground-truth SVGs for code efficiency calculation. Resolve this inconsistency to avoid reader confusion.

**Ethical Concerns:**

["NO or VERY MINOR ethics concerns only"]

**Final Justification:**

The idea of this paper is simple and clear, and the experimental results demonstrate its effectiveness. Therefore, I initially gave it a high score. I appreciate the authors‘ rebuttal and will keep my recommendation unchanged.

**Limitations:**

Yes

**Paper Formatting Concerns:**

Please refer to the point 2 in Questions

**Quality:**

4

**Strengths And Weaknesses:**

Strength:
1. The motivation is clear and easy to grasp, and its distinction from existing work is well articulated (Lines 39–46).
2. The method achieves new state-of-the-art results on task metrics while demonstrating a significant improvement in code efficiency.
Weakness:
Please find details in Questions.

In my opinion, this is a well-matured paper with clear motivation and solid results. While the key insight seems simple, the paper demonstrates high completeness and sufficient effort throughout. All the experimental results I care about are provided, albeit mostly in the supplementary material due to space constraints. Therefore, I give an **Accept** rating, but I encourage the authors to refine the content further based on the feedback in the Questions.

---

> ### Author Rebuttal · Authors · 2025-07-31
>
> We sincerely thank the reviewer for the thoughtful and positive feedback. We are glad that the motivation and distinction from prior work came through clearly, and we appreciate your recognition of the state-of-the-art results and the overall high completeness of our paper.
>
> We are especially encouraged by your appreciation of the *simplicity* of our approach. We believe the best ideas are often the simplest ones: easy to understand, straightforward to implement, yet capable of delivering strong and generalizable results. This principle guided our design of RLVG.
>
> Thanks for your questions and feedback for improving our manuscript's clarity. We have applied all your concerns as follows:
>
> -----------------
>
> ## **Q1. Terminology: “inverse rendering code generation problem” and “perceptual inputs”**
>
> Thank you for pointing this out. **We agree that the term “inverse rendering code generation problem” is not widely adopted** and could be confusing. We have removed the phrase “known as” and replaced it with a clearer and more intuitive description: **visual code generation**. Additionally, **we have replaced the term “perceptual inputs” with more precise wording** (“images or textual descriptions”).
>
> The revised opening sentence of the introduction now reads:
> *“Visual Code Generation is the problem of translating images or textual descriptions into executable code that gets compiled and produces a visual output [Rodriguez et al., 2025b; Baulé et al., 2021].”*
>
> For completeness, the previous version was:
> *“Generating structured visual code from perceptual inputs, known as the inverse rendering code generation problem, aims to translate images or text into executable code that reproduces the target visual content [Rodriguez et al., 2025b; Baulé et al., 2021].”*
>
> We have also ensured this terminology is used consistently throughout the paper. Please let us know if you have further suggestions.
>
> --------
>
> ## **Q2. Citation style: switch to numeric to free space to move Appendix C.2**
>
> Thank you for the suggestion. We agree that several important results were previously buried in the supplementary. Following your advice, **we have switched to a numeric citation format to save space** and we have moved key content into the main paper.
>
> Specifically, the reward details from Appendix C.2, along with select results from Appendices A.1, A.2, and A.3, are now integrated into the main experimental sections.
>
> --------------
>
> ## **Q3. Figure 1 inconsistencies**
>
> Thank you for pointing out these issues with our main figure. Your feedback has allowed us to improve its clarity.
> **We have applied your changes as follows:**
>
> ### **Q3.a) Caption clarity regarding inputs**
>
> You are right, the SVG is not an input to the model, and the wording was misleading. We intended to convey that the model's outputs are compared to *task targets*: the input image, the input text, or the ground truth SVG (if available). **We have revised the caption to remove the ambiguous reference to "inputs"** and clarified what the comparisons are made against.
>
> **The updated caption now reads:**
>
>  *“Given a text or image input, the model generates multiple SVG rollouts, which are rendered into images to compute a reward based on the task at hand using the image or text targets, or the ground truth SVG (if available).”*
>
> Previously, it read:
>
>  *“Given a text or image input, the model generates multiple SVG rollouts, which are rendered and compared to the input (image, SVG, or text) to compute rewards based on reconstruction, semantics, and code efficiency.”*
>
> We believe this change makes the caption more precise and consistent with the description in the introduction.
>
> -------------
>
> ### **Q3.b) Missing legend in Figure 1**
>
> We have **added a small legend in the top-right corner of Figure 1** to clarify the meaning of the stop-sign icon, which denotes non-differentiability in the rendering pipeline.
>
> --------
>
> ### **Q3.c) Clarifying ground-truth SVG availability for the example shown**
>
> Thank you for catching this ambiguity. **The “apple” example shown in Figure 1 is from the inference (test) phase**, not from the training set. This specific sample is used to visualize how the model's outputs evolve *over the course of RL training*, and no rewards are computed on it.
>
> We agree that the confusion arises because Figure 1 appears alongside our explanation of the RL training loop, which does involve reward computation using image, text, or SVG targets, depending on the task. However, the visualized sample is only used for monitoring progress on a fixed out-of-distribution Im2SVG test example, for which we only have the raster image, and no ground-truth SVG is required or used.
>
> **We have now clarified this explicitly in the caption**, and it will be visible during camera-ready, to distinguish between the general RL training setup and the illustrative test sample.
>
> Thanks again for your review

---

> > ### Comment · Reviewer_RqsR · 2025-08-04
> >
> > Thanks for the authors' rebuttal. I'm glad to keep my positive score.

---

### Official Review · Reviewer_Lhdn · 2025-07-02

**Clarity:** 4
**Significance:** 3
**Originality:** 3
**Rating:** 5
**Confidence:** 3

**Summary:**

This article introduced a reinforcement learning based approach for SVG generation with autoregressive vision-language models. Also, the authors provide a set of rewards for Vector Graphics Rendering in pixel-level, semantic and simplicity to solve the non-differentiable rendering problem.

**Questions:**

Should these two tasks(Im2SVG and Text2SVG) be done in same model? Is it possible to finetune these two model separately for industrial purposes? For the Im2SVG model, more comparison such as Im2VEC need to be done, not limited in VLM models.

**Ethical Concerns:**

["NO or VERY MINOR ethics concerns only"]

**Final Justification:**

Experiments in rebuttal partly solved most of my questions. Q4 is partly solved while others(Q1, Q2 and Q3) completely solved. Experiment 1 proved that Canny-L2 does work well. Experiment 3 added other models comparison. I've updated my score in final decision.

**Limitations:**

Results on binary images vectorization? Can this method keep topological and back-front logical correct?

**Quality:**

4

**Strengths And Weaknesses:**

Strength: Use SVG-SFT and RL to solve the problem. Adequate experiments, especially in the ablation studies are persuasive.

Weakness: I wondering simply use L2 reward is a good idea in Eq.(5). In image generation, L2 Loss function may lead to blur or tropology-misleading results, maybe L1 or other reward functions should also be tried. Also, the canny edge detector based L2 variant should be further discussed.

---

> ### Author Rebuttal · Authors · 2025-07-31
>
> Thank you for the constructive review and for highlighting the strengths of our approach, such as the clear motivation for combining SVG‑SFT and RL to address non‑differentiable rendering and the thorough ablation studies. Your feedback has greatly improved the quality of our paper.
>
> Below, we address every point you raised and indicate the concrete additions we have made in the revised manuscript, which will be visible in the camera-ready version.
>
> -------
>
> ## **Q1  “Simply using L2 may blur shapes; try other pixel rewards like L1.”**
>
> **We don’t simply use L2**. As shown in the “Final Reward Aggregation” Section 3.2 (line 186), we use a *combination of rewards*. **We realized that the exact weights were not specified clearly in the main paper, and we have now included them**. The best performing combination includes **L2, Canny-L2, and LPIPS, with a small code length deviation term**. As explained in Section 5.2, a *higher L2 weight leads to better vectorization accuracy*, while a *higher LPIPS weight pushes more toward style preservation*.
>
> We’ve updated the main text for the camera-ready version to make these trade-offs and choices more explicit.
>
> **Experiment**: Following your suggestion, **we also experimented with additional rewards to further analyze their impact**. We conducted a new ablation comparing L1, L2, Canny-L2, and SSIM as pixel-level losses.
>
> **Setup**: We experiment on a 7B Queen 2.5VL model that has completed 5 epochs of SVG-SFT. We train with RLVG with our most optimized RL (GRPO) setup for 240 steps, with 32 samples per step and 32 rollouts (7k train samples in total), using 16 H100 GPUs over 48 hours.
>
> | Method   | Code Eff. | DinoScore | L2     | LPIPS  | SSIM   |
> |----------|-----------|-----------|--------|--------|--------|
> | L2 Canny | -334      | 98.70     | 1.03   | 3.08   | 95.08  |
> | L2       | -159      | 98.82     | 1.10   | 3.17   | 94.76  |
> | SSIM     | -210      | 98.64     | 1.30   | 3.34   | 94.76  |
> | L1       | -254      | 98.69     | 1.57   | 4.06   | 93.62  |
>
>
> We have added this table and a qualitative figure to compare visually, and both will be available during camera-ready. We note the following findings.
>
> **Main findings:**
>
> - **Finding 1**:  **L2 Canny is the best performant**. As seen in the table above, L2 Canny gives the best performance, though L2 is pretty similar. **L1 gives the world results falling by 0.004 points over the best**. In qualitative terms, we see the *L2 Canny helps get the main shapes and edges well placed, though it fails at very small patterns and details*, *L2 is better here at smaller details*. As we show in the paper, **a combination of rewards is the best approach**.
>
> We find it interesting to point out that optimizing for SSIM does not make it reach the best SSIM at test, so no reward hacking happens. L2 Canny is the best performance on SSIM as well.
>
> - **Finding 2**: **No blurring**. When using L2 alone, *we do not observe the typical blurring artifacts often seen in raster image generation*. This is likely because SVGs are rendered from discrete and **sharp primitives, and the output space is inherently crisp and well-structured**. However, we do observe that L2 alone can lead to artifacts such as over-closed paths or small geometric misalignments, especially in cases involving thin strokes or disconnected segments. As shown before, L2 Canny provides a better signal for getting sharp shapes and good edges
>
> ----------------------
>
> ## **Q1.1: The Canny edge detector-based L2 variant should be further discussed.**
>
> Thank you for pointing this out. We now provide a clearer explanation of the **Canny-L2** reward, which was not sufficiently detailed in the original submission.
>
> This rewards applies a **Canny edge detector** to both the input and predicted images, followed by optional **dilation** and **Gaussian blur**, before computing the L2 loss. We use the `cv2.Canny` implementation from OpenCV, followed by `cv2.dilate` with a configurable kernel and number of iterations.
>
> Specifically:
>
> - We use `dilate=True` to expand detected edges and improve alignment tolerance.
> - The dilation uses a square kernel of size `dilate_kernel_size=3`, applied for `dilate_iterations=1`.
> - After preprocessing, both edge maps are normalized (zero mean, unit variance), and we compute the clipped L2 loss.
>
> This formulation encourages the model to match structural boundaries more precisely, especially in settings with thin strokes, closed shapes, or topology-sensitive inputs.
>
> **Results**: As shown in the ablation table above, **Canny-L2 outperforms standard L2 by 0.001 in MSE and 0.01 in SSIM**, confirming its benefit for structural alignment.
>
> ----------
>
> ## **Q2: Should Im2SVG and Text2SVG live in the same model?**
>
> **Yes, a single model can be trained jointly on both Image2SVG and Text2SVG** tasks by constructing appropriate conversation templates and leveraging datasets that include textual annotations.
>
> **Experiment**: We tested this with the Qwen2.5VL‑3B model using the SVG‑Stack dataset, which contains both image and text prompts. Training was run on 32 × H100 GPUs for 24 hours, a single model can be trained jointly on both Image2SVG and Text2SVG tasks by constructing appropriate conversation templates and leveraging datasets that include textual annotations. The model successfully learned to handle both modalities.
>
> For comparison:
> - A model trained only on Image2SVG achieves **MSE = 0.04**.
> - A model trained on both tasks jointly reaches **MSE = 0.07**.
>
> We note that the joint model underperforms slightly in Image2SVG compared to a task-specific one, likely due to increased task complexity and insufficient training time. We are currently running a longer training run that matches the number of image samples used in the standalone Image2SVG training, and we will report those results in the camera-ready version. This could yield a stronger foundation model capable of handling both tasks effectively. That said, we still foresee value in deploying dedicated fine-tuned models for Image2SVG when the goal is high-precision vectorization, as specialization may offer better performance.
>
> ---------------------
>
> ## **Q3: For the Im2SVG model, more comparisons such as Im2VEC should be included, not limited to VLM models.**
>
> We did not limit our comparisons to VLM-based models. As detailed in Section 4.2, we already included:
>
> - **Three classic image processing methods**: VTracer, PoTracer, and PyAutoTrace
>
> - **Two parametric deep learning approaches**: DiffVG and LIVE
>
> These baselines represent the strongest alternatives to autoregressive VLMs across image processing, differentiable rendering, and deep learning methods.
>
> **Regarding Im2VEC**, we had previously tested it and found it to be highly overfitted to small-scale emoji datasets. In practice, we found it **infeasible to retrain effectively** on the large and diverse SVG-Stack dataset due to poor generalization. Nonetheless, **we’ve now included its performance, as well as an additional recent baseline SuperSVG**, which was suggested by Reviewer STcx.
>
> We summarize the results in the table below:
>
> | Model           | MSE ↓ | SSIM ↑ | LPIPS ↓ | DINO ↓ | Code Eff. |
> |-----------------|-------|--------|---------|--------|----------------|
> | PoTracer        | 8.15  | 77.28  | 89.23   | 19.10  | -7.3k          |
> | DiffVG          | 6.64  | 81.23  | 86.12   | 20.5   | -19.7k         |
> | PyAutoTrace     | 4.71  | 87.44  | 95.68   | 10.71  | -99.7k         |
> | VTracer         | 4.25  | 87.94  | 95.75   | 11.66  | -12.9k         |
> | LIVE            | 2.22  | 88.11  | 93.45   | 7.23   | -18.3k         |
> | **SuperSVG**    | 3.05  | 83.3   | 13.5    | 82.7   | -65.6k         |
> | **StarVector**  | 3.46  | 88.0   | 98.0    | 7.51   | -127           |
> | **Im2VEC**      | **18.1**  | **76.5**   | **29.1**    | **69.2**   | **-4.3k**          |
> | **RLVG (ours)** | **1.03**  | **95.1**   | **3.08**    | **98.7**   | **-334**           |
>
> We have included the most up-to-date results for RLVG performed in this rebuttal. As seen above, our method outperforms all baselines across all metrics, including those not based on vision-language pretraining.
>
> ---------------------
>
> ## **Q4: Can the model accurately vectorize binary images and preserve topology and layer order?**
>
> This is a great question. To evaluate this, **we created a small test set** composed of 50 SVGs by extracting a subset of black-and-white examples from SVG-Stack. We selected SVGs composed of primitives such as circles, polygons, and basic shapes, and binarized them by setting all vector colors to black over a white background.
>
> **Results**: Visually, the resulting rendered images generated by our method remain crisp, and the SVGs generated by our model correctly utilize primitives more effectively than traditional binary-focused methods like Potrace (only using path). We include preliminary results below:
>
> | Model          | MSE ↓ | SSIM ↑ | LPIPS ↓ | DINO ↓ | Code Eff. |
> |----------------|-------|--------|---------|--------|----------------|
> | Potrace        | 8.53   | 77.02   |   19.23  | 90.21   | -7.3k          |
> | VTracer        | 4.32   | 88.01   |  11.51   | 96.21   | -12.9k         |
> | **RLVG (ours)**| **1.11** | **96.22** | **3.32**   | **98.52** | **-334**         |
>
>
> Interestingly, *we find that this task is easier for RLVG* (and for VLM SVG models in general), as it removes many of the challenges related to color, shading, and texture. The model focuses purely on geometric structure, and we observe high-quality outputs that preserve the correct topology and element layering.
>
> We have included this analysis in our manuscript and will be available in the camera-ready version.
>
> Thanks again for your review

---

> ### Comment · Reviewer_Lhdn · 2025-08-02
>
> Thank you again for your rebuttal and experiments, and I'm looking forward to your camera ready version.
>
> Two small questions about Q4:
> 1. How to evaluate "topology correct"? Previous image-based evaluations may not be able to work in this area.
> 2. Have you ever tried complex binary graphs such as paper cuts, maps or character images?
>
>
> P.S. some bugfix in your rebuttal: The table in Q1 section requires upper and lower arrows.

---

> ### Author Response · Authors · 2025-08-02
> **Cool Binary Image Topology Use-Case**
>
> Thanks for your response and follow-up questions. Below, we address your latest question on Q4:
>
> ### **1. How do you evaluate "topology correctness"? Prior image-based evaluations may not apply here.**
>
> You're absolutely right, *traditional image-based metrics primarily focus on pixel-level or perceptual similarity*, which often overlook whether the generated SVGs preserve the underlying topological structure of the image (e.g., shapes and higher-order primitives).
>
> To better evaluate this aspect, we propose the following approach:
>
> **Qualitative Evaluation**: We inspect both the SVG code and its rendered output to determine whether the model correctly uses SVG primitives (e.g., `<circle>`, `<rect>`, `<polygon>`) in place of general-purpose `<path>` elements when the shape is visually recognizable.
>
> **Quantitative Evaluation (Primitive Usage Metric)**:
> Using the 50-sample test set described in Q4 (containing recognizable primitives), we compute the proportion of samples where the model correctly uses at least one SVG primitive (other than `<path>`).
>
> - A score of **100%** means the model used the correct primitives in all 50 samples.
>
> - A score of **0%** indicates it relied solely on paths, failing to recognize any higher-level structure.
>
> In *qualitative terms*, models like StarVector, SVG-SFT, and RLVG can often **detect and generate meaningful primitives** (e.g., circles, rectangles) when the shape is clearly distinguishable. RLVG improves over SVG-SFT in this regard, and we believe further gains are possible with rewards that explicitly encourage primitive use (left for future work). In contrast, *prior models like Potrace, VTracer, LIVE, and SuperSVG are fundamentally limited and do not capture topological structure*.
>
> **Data quality plays a key role**: if training SVGs use only `<path>` elements, models will fail to learn higher-level structure. Future work could explore simplifying such SVGs by converting paths into primitives.
>
> In *quantitative* terms, our primitive-usage metric shows:
>
> - RLVG: **76%**
> - SVG-SFT: **54%**
> - Image Processing methods: **0%** (limited by design)
>
> ### **2. Have you ever tried complex binary graphs such as paper cuts, maps or character images?**
>
> Thanks for the suggestion. While we haven’t explicitly evaluated on **paper cuts or maps**, the SVG-Stack dataset includes some samples of this kind due to its raw and large-scale nature. Thanks to your suggestion, **we plan to construct targeted test sets for these use cases to better assess model performance under such conditions**.
>
> As for **character images**, we experimented with the **SVG-Fonts dataset** [1], which contains single-character, black-filled SVGs across various fonts. As shown in Table 3 of the supplementary material, **RLVG performs well on this dataset without any finetuning**, demonstrating **strong generalization**. However, since SVG-Fonts mostly encode characters as `<path>` elements, the output does not preserve explicit topology or primitive structure, reflecting the nature of the data itself [1].
>
> [1] Rodriguez, Juan A., et al. "Starvector: Generating scalable vector graphics code from images." arXiv preprint arXiv:2312.11556 (2023).
>
> ### **P.S. some bugfix in your rebuttal: The table in Q1 section requires upper and lower arrows.**
>
> Thanks for catching that! We’ve updated the table to include the correct ↑ and ↓ indicators, as shown below.
>
> | Method   | Code Eff. ↓ | DinoScore ↑ | L2 ↓   | LPIPS ↓ | SSIM ↑ |
> |----------|--------------|-------------|--------|---------|--------|
> | L2 Canny | -334         | 0.9870      | 0.0103 | 0.0308  | 0.9508 |
> | L2       | -159         | 0.9882      | 0.0110 | 0.0317  | 0.9476 |
> | SSIM     | -210         | 0.9864      | 0.0130 | 0.0334  | 0.9476 |
> | L1       | -254         | 0.9869      | 0.0157 | 0.0406  | 0.9362 |
>
>
> ### **Thanks!**
>
> We appreciate your prompt engagement and your suggestion regarding binary images, paper cuts, and topology-aware character models. This is an important use case, and we plan to initiate a dedicated research thread to support it.
>
> If our responses have addressed your concerns, we would be grateful if you could consider updating your score to better reflect your current view.

---

> > ### Comment · Reviewer_Lhdn · 2025-08-06
> >
> > I thank the authors for the rebuttal. I believe this is good work with adequate experiments, and please remember to add above experimental results in final version.

---

### Author Response · Authors · 2025-08-09
**General Comment**

We thank all reviewers for their thoughtful reviews, constructive questions, and active engagement. **Your feedback, proposed experiments, and requests for clarification have greatly improved our paper**, and we will **integrate all comments into the camera-ready version**. We are pleased by the overall positive assessment and glad that our responses addressed the main concerns, leading all reviewers to accept the paper.

### **Main points addressed:**

- **Novelty & Contribution**: Clarified our methodological advances beyond StarVector and positioned RLVG as the first rendering-aware RL framework for SVG generation with autoregressive VLMs.

- **Reward Functions**: Expanded explanation and ablations for L1, L2, Canny-L2, and SSIM; clarified weight choices and impact on performance.

- **Qualitative & Code Comparisons**: Added side-by-side visual and code examples (for camera ready) for StarVector vs. StarVector+RLVG, and described improvements in structure, alignment, and efficiency.

- **Additional Baselines**: Included SuperSVG and LLM4SVG, with results integrated into main tables.

- **User Study**: Conducted a study with designers and SVG experts confirming RLVG’s advantages in both fidelity and editability over SFT.

- **Topology Evaluation**: Detailed methodology for assessing topology correctness in binary images and reported results.

- **Compute Cost**: Discussed strategies to reduce RL training cost, including roll-out reductions, distillation, and off-policy updates.

- **Broader Impact**: Shared preliminary results for applying RLVG to HTML/CSS and Tikz and noted relevance to other rendering-aware code generation tasks.

We appreciate the constructive dialogue and look forward to presenting the improved version of this work.

---

### Note · Authors · 2025-08-12

**Please see our general comment below.**

Thank you all again for an excellent and engaging review process.

RLVG Authors

---

### Decision · Program_Chairs · 2025-09-17

**Decision:**

Accept (poster)

**Comment:**

The paper received four expert reviews. The authors provided a rebuttal that attempted to address the concerns raised in the reviews. The reviewers read the rebuttal and engaged with the authors.  The reviewers unanimously like the paper and recommended accept. The area chair agreed with the recommendation and decided to accept the paper. Congratulations! Please see the reviews for feedback on the paper to revise the final version of your paper and include any items promised in your rebuttal.